# Unconstrained Salient and Camouflaged Object Detection

## Abstract

Visual Salient Object Detection (SOD) and Camouflaged Object Detection (COD) are two interrelated yet distinct tasks. Both tasks model the human visual system's ability to perceive the presence of objects. The traditional SOD datasets and methods are designed for scenes where only salient objects are present, similarly, COD datasets and methods are designed for scenes where only camouflaged objects are present. However, scenes where both salient and camouflaged objects coexist, or where neither is present, are not considered. This simplifies the existing research on SOD and COD. In this paper, to explore a more generalized approach to SOD and COD, we introduce a benchmark called Unconstrained Salient and Camouflaged Object Detection (**USCOD**), which supports the simultaneous detection of salient and camouflaged objects in unconstrained scenes, regardless of their presence. Towards this, we construct a large-scale dataset, **CS12K**, that encompasses a variety of scenes, including four distinct types: scenes containing only salient objects, scenes with only camouflaged objects, scenes where both salient and camouflaged objects coexist, and scenes without any objects. In our benchmark experiments, we find that a major challenge in USCOD is distinguishing salient objects from camouflaged objects within the same model. To address this, we propose a USCOD baseline called **USCNet**, which freezes the SAM mask decoder for mask reconstruction, allowing the model to focus on distinguishing between salient and camouflaged objects. Furthermore, to evaluate models' ability to distinguish between salient and camouflaged objects, we design a metric called Camouflage-Saliency Confusion Score (**CSCS**). The proposed method achieves state-of-the-art performance on the newly introduced USCOD task. The code and dataset will be publicly available.

## 1 Introduction

The attention mechanism is one of the key cognitive functions of humans Posner et al. (1990). In real-world scenarios, people are often drawn to salient objects while overlooking camouflaged ones. The goal of Salient Object Detection (SOD) is to detect objects in an image that the human visual system considers most salient or attention-grabbing, while Camouflaged Object Detection (COD) aims to detect objects that are difficult to perceive or blend seamlessly with their surroundings Li et al. (2021). SOD simulates the human ability to focus on salient objects, while COD mimics the human ability to discover camouflaged objects. Both of them exhibit significant potential across various fields, such as anomaly detection in medical image analysis Tang et al. (2023), obstacle recognition in autonomous driving, camouflage detection in military reconnaissance Lin & Prasetyo (2019), and wildlife tracking in environmental monitoring Stevens & Merilaita (2009). Currently, existing methods follow the training and inference paradigms of popular datasets, such as COD10K Fan et al. (2020) in COD and DUTS Wang et al. (2017a) in SOD, and have made significant progress.

**Limitations of existing SOD and COD methods.** Existing SOD and COD methods often rely on strong pre-defined constraints specific to the tasks, which may limit their generalizability. The classic SOD and COD methods are designed for detecting their respective attribute-specific objects, considering only scenes where single-attribute objects exist (Figure 1. Scene A and Scene B). They overlook more complex scenes where both salient and camouflaged objects coexist (Figure 1. Scene C) or where neither type of object is present (Figure 1. Scene D). Some works have already explored how to handle SOD and COD simultaneously. EVP Liu et al. (2023) achieves the detection of

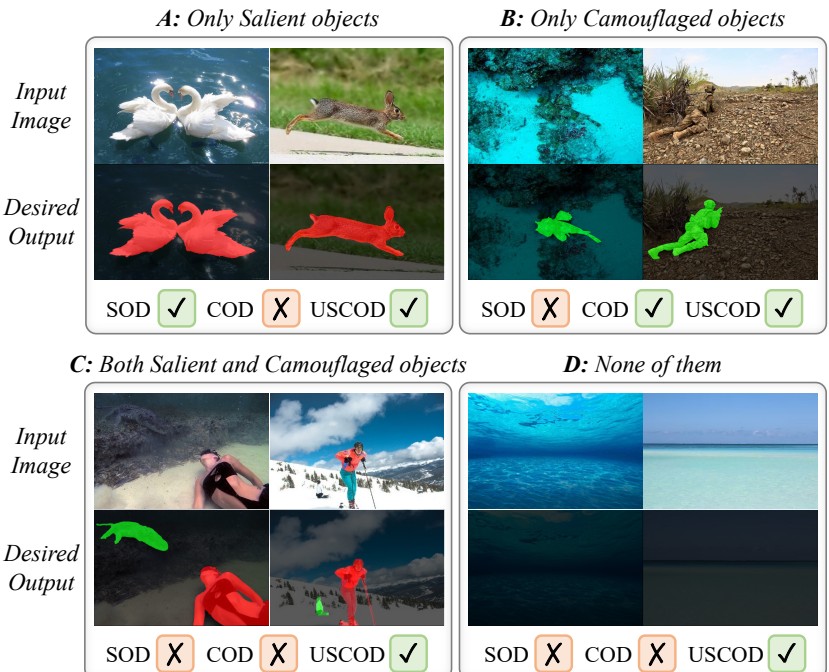

Figure 1: SOD supports scenes that only exist salient objects, *e.g., (A)*. COD supports scenes that only exist camouflaged objects, *e.g., (B)*. Compared with classic SOD and COD, the proposed **Unconstrained Salient and Camouflaged Object Detection (USCOD)** supports scenes may existing salient objects, camouflaged objects, both, or neither, *e.g., (A-D)*. In Desired Output, the **red** mask indicates the salient object, and the **green** mask represents the camouflaged object.

salient and camouflaged objects by switching different visual prompts. VSCode Luo et al. (2024) and Spider Zhao et al. (2024) achieve the detection of salient and camouflaged objects through joint training on multiple datasets and specific task prompts. However, these methods have certain constraints. The visual prompt of EVP needs to be retrained according to the datasets of different tasks and requires pre-defining the category of the detection task. Similarly, in VSCode and Spider, while only one combined training process is needed for multiple datasets of different tasks, the task prompts for difference datasets still need to be pre-defined. These methods cannot adaptively detect salient and camouflaged objects based on the content of the image. Furthermore, these methods cannot handle situations where both salient and camouflaged objects exist in the same image.

**New benchmark and dataset.** To overcome these constraints, we propose a new benchmark called Unconstrained Salient and Camouflaged Object Detection (USCOD), which allows the detection of both salient and camouflaged objects in unconstrained scenes (refer to Figure 1. Scene A-D). However, most existing SOD and COD datasets do not include Scene C and Scene D, as they only contain scenes with single-attribute objects. Although COD10K has collected image samples for Scene D, these samples have not yet been effectively utilized in models. To address the limitations of existing datasets and advance USCOD research, we construct a new USCOD dataset named CS12K. It contains 12,000 images, covering the four scenes: 3,000 images of Scene A; 3,000 images of Scene B; and two scenes lacking in existing datasets, including 3,000 images of Scene C and 3,000 images of Scene D, which are manually collected and annotated. The comparison of the data analysis between our dataset and the existing SOD and COD datasets is shown in Table 1.

**New evaluation metric.** For the USCOD benchmark, one key issue is how to evaluate the ability of the model to understand the semantic differences between salient and camouflaged objects. However, existing metrics fail to effectively capture this ability, as they only assess the detection performance of salient and camouflaged objects individually, such as weighted F-measure Margolin et al. (2014), Structural measure Fan et al. (2017). To fill this gap, we design a metric called

Camouflage-Saliency Confusion Score (CSCS) to evaluate the ability of the model to distinguish between salient and camouflaged objects.

**Challenge and A baseline method.** To explore solutions for the USCOD problem, we retrain and evaluate 19 SOD and COD models. Our findings reveal that existing models struggle to accurately distinguish between salient and camouflaged objects in unconstrained scenes, often leading to confusion. For example, in Scene A of Figure 5, a prominent duck may be misidentified as a camouflaged object, while in Scene C of Figure 5, a person disguised as grass is recognized as a salient object. To address this issue, we propose a USCOD baseline model, USCNet, which decouples the learning of attribute distinction from mask reconstruction. By freezing the SAM mask decoder, allowing it to focus on attribute distinction of salient objects, camouflaged objects, and background. Additionally, we design an APG module that integrates dynamic and static queries to enhance the semantic differentiation between salient and camouflaged objects. The results demonstrate that decoupling the learning processes enables USCNet to achieve state-of-the-art performance across all metrics in overall scenes, *e.g.*, 78.03% on the mIoU and 7.49% on the CSCS.

In summary, our contributions are listed as follows:

- We propose a new benchmark called **USCOD**, which supports the detection of both salient and camouflaged objects in unconstrained scenes. Further, a new metric, **CSCS**, is introduced to assess the model's confusion between salient and camouflaged objects.

- We introduce a large-scale USCOD dataset **CS12K**. To our knowledge, this is the first dataset that covers multiple scenes without restrictions on the presence of salient or camouflaged objects.

- A novel baseline **USCNet** decouples the learning of attribute distinction from mask reconstruction, utilizing an Attribute-specific Prompt Generation (APG) that focuses on differentiating salient objects from camouflaged objects, while the frozen SAM mask decoder is used for reconstructing the object masks.

- Based on CS12K, we establish the complete CS12K **benchmark** to conduct a broader study of the USCOD task. USCNet is compared with 19 cutting-edge SOD and COD models and shows promising performance.

## 2 RELATED WORK

### 2.1 SALIENT AND CAMOUFLAGED OBJECT DETECTION

**SOD.** In recent years, salient object detection models have focused on better detecting salient objects in images using various approaches. The main approaches can be divided into attention-based methods Liu et al. (2018); Piao et al. (2019); Zhang et al. (2018), multi-level feature-based methods Fang et al. (2022); Hou et al. (2017); Pang et al. (2020); Wang et al. (2017b); Zhao et al. (2020), and recurrent-based methods Deng et al. (2018); Liu & Han (2016); Wang et al. (2018). Saliency detection Zhao et al. (2020); Zhang et al. (2021); Liu et al. (2021a); Zhuge et al. (2022) primarily focus on achieving saliency predictions while preserving the structure.

**COD.** Compared to salient object detection, current COD methods Fan et al. (2021); Mei et al. (2021); Pang et al. (2022); He et al. (2023); Jia et al. (2022) focus primarily on edge-aware perception and texture perception. Mainly divided into the following two types, multi-level feature-based methods Zhang et al. (2022); Yang et al. (2021); Ren et al. (2021); Zhai et al. (2022), Edge joint learning Zhai et al. (2021); Sun et al. (2022); He et al. (2023).

**Unified.** SOD and COD are distinct yet interrelated tasks Luo et al. (2024). Recently, some works have already begun to unify the two tasks. EVP Liu et al. (2023) solves the detection of Camouflaged, Forgery, Shadow, and Defocus Blur by adding a visual prompt to the same base segmentation model, allowing a single base model to handle different tasks by using specific visual prompt. VSCode Luo et al. (2024) uses a multi-dataset joint training approach, simultaneously utilizing datasets from RGB SOD, RGB COD, RGB-D SOD, RGB-D COD, RGB-T SOD, Video SOD (VSOD), and VCOD, and assigning different task prompts for SOD and COD tasks to achieve task unification. Similarly, Spider Zhao et al. (2024) uses a comparable method to unify Context-dependent tasks. It also requires the input of specific task prompts.

The unified models mentioned above have two constrains. First, they require the task type to be pre-defined in advance, with a specific prompt input into the model, thus being constrained by the prompt. Second, they cannot handle scenarios where both salient object and camouflaged object are present simultaneously, being constrained by the scenario. The USCOD task we proposed successfully addresses these two aspects, achieving unconstrained by both the prompt and the scenario.

## 2.2 APPLICATIONS OF SAM.

The Segment Anything Model (SAM) Kirillov et al. (2023) represents a significant advancement in scene segmentation using large vision models. Its versatility and adaptability underscore its capability to comprehend complex scenarios and objects, thereby pushing the boundaries of image segmentation tasks even further. Current works leveraging SAM Chen et al. (2023a); Zhang et al. (2023); Xiong et al. (2023); Ma et al. (2024) showcase its adaptability to downstream tasks, notably in areas where traditional segmentation models struggle, such as EfficientSAM Xiong et al. (2023) and MedSAM Ma et al. (2024). More recently, the release of SAM2 Ravi et al. (2024) enhances the original SAM's ability to handle video content while demonstrating improved segmentation accuracy and inference efficiency in image segmentation across various downstream applications Zhu et al. (2024); Yan et al. (2024); Lian & Li (2024); Lou et al. (2024).

Some works that use SAM for SOD and COD are closely related to our research. MDSAM Gao et al. (2024) is a novel multi-scale and detail-enhanced SOD model based on SAM, aimed at improving the performance and generalization capability of SOD task. SAM-Adapter Chen et al. (2023a) and SAM2-Adapter Chen et al. (2024) offers a parameter-efficient fine-tuning way to enhance performance of SAM and SAM2 in downstream tasks like COD and medical image segmentation by adding task-specific knowledge. Nevertheless, these methods may be suboptimal for fine-tuning SAM for the USCOD task, as they discard prompt architecture of SAM and tune the mask decoder to simultaneously learn distinguishing attributes and segmenting mask, even though the mask decoder is not designed for attribute distinction. Therefore, we retain the mask decoder solely for mask reconstruction and use independent learning for attribute distinction to better differentiate between salient objects, camouflaged objects, and background.

## 3 PROPOSED CS12K DATASET

The current datasets for camouflaged object detection, such as COD10K Fan et al. (2020), CAMO Le et al. (2019), NC4K Lv et al. (2021), primarily feature scenes with exclusively camouflaged objects. Similarly, datasets for salient object detection, such as DUTS Wang et al. (2017a), and HKU-IS Li & Yu (2015), predominantly focus on scenes with solely salient objects. There are relatively few samples with both salient objects and camouflaged objects in an image, which is not conducive to the realization of the unconstrained existence of salient and camouflaged object detection. Therefore, we introduce the **CS12K**, a dataset that includes more comprehensive and complex scenarios for unconstrained salient and camouflaged object detection. It includes scenes with both salient and camouflaged objects, scenes with only one type, and scenes without either. We will describe the details of CS12K in terms of three key aspects, as follows.

### 3.1 DATA COLLECTION

Under the premise of ensuring sample balance, we collect 12,000 images from 8 different sources and divide them into four scenes after manual filtering: (A) Scenes with only salient objects: 3,000 images containing only salient objects selected from SOD datasets DUTS and HKU-IS; (B) Scenes with only camouflaged objects: 3,000 images containing only camouflaged objects selected from COD datasets COD10K and CAMO; (C) Scenes with both salient and camouflaged objects: 342 images from the COD datasets COD10K, CAMO, and NC4K, along with 41 images from the datasets LSUI Peng et al. (2023) and AWA2 Xian et al. (2018), and an additional 2,617 images collected from the internet, making a total of 3,000 images; (D)Scenes without salient and camouflaged objects, considered as background: 1,564 images from COD10K, and 1,436 images from the Internet. Finally, we get 12,000 images, with the training set containing 8,400 images and the testing set containing 3,600 images. The data source is shown in Figure 2 (Left).

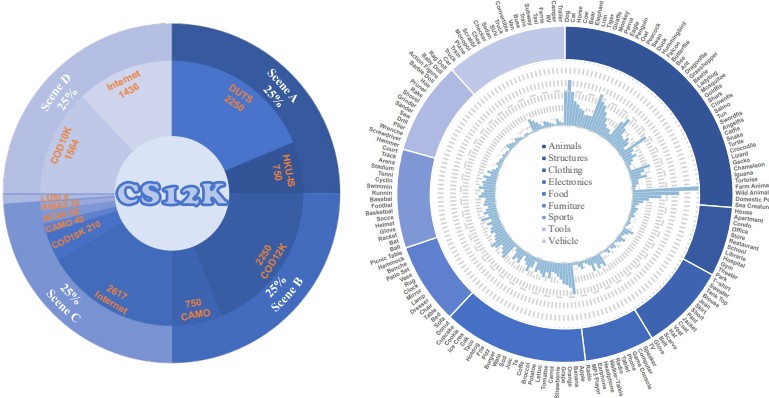

Figure 2: **Left**: The data source and distribution of different data types. **Right**: Categories and groups of our CS12K dataset. **Zoom-in for better view**.

Table 1: Data analysis of existing datasets.

| Task | Dataset | #Ann. IMG | Class | Scene A | Scene B | Scene C | Scene D |
|------|---------|-----------|-------|---------|---------|---------|---------|
| SOD | SOD Movahedi & Elder (2010) | 300 | - | 300 | ✗ | ✗ | ✗ |
| | PASCAL-S Li et al. (2014) | 850 | - | 850 | ✗ | ✗ | ✗ |
| | ECSSD Yan et al. (2013) | 1000 | - | 1000 | ✗ | ✗ | ✗ |
| | HKU-IS Li & Yu (2015) | 4447 | - | 4447 | ✗ | ✗ | ✗ |
| | MSRA-B Liu et al. (2011) | 5000 | - | 5000 | ✗ | ✗ | ✗ |
| | DUT-OMRON Yang et al. (2013) | 5168 | - | 5168 | ✗ | ✗ | ✗ |
| | MSRA10K Cheng et al. (2015) | 10000 | - | 10000 | ✗ | ✗ | ✗ |
| | DUTS Wang et al. (2017a) | 15572 | - | 15572 | ✗ | ✗ | ✗ |
| | SOC Fan et al. (2018a) | 3000 | 80 | 3000 | ✗ | ✗ | ✗ |
| COD | CAMO Le et al. (2019) | 1250 | 8 | ✗ | 1250 | ✗ | ✗ |
| | CHAMELEON Skurowski et al. (2018) | 76 | - | ✗ | 76 | ✗ | ✗ |
| | NC4K Lv et al. (2021) | 4121 | - | ✗ | 4121 | ✗ | ✗ |
| | COD10K Fan et al. (2020) | 7000 | 78 | ✗ | 5066 | ✗ | 1934 |
| USCOD | **CS12K(Ours)** | 12000 | 179 | 3000 | 3000 | 3000 | 3000 |

## 3.2 DATA ANNOTATION

We use SAM Kirillov et al. (2023) for mask labeling and manual correction. When labeling, we first retain the RGB pixels corresponding to the object instance in the image, set the remaining pixels to 0, obtain the rough classification results through CLIP Radford et al. (2021), and then perform manual comparison and correction. In addition to the camouflaged object category labels already included in the images from COD10K Fan et al. (2020), the remaining objects require category assignment. Some example images of different scenes from our CS12K dataset are shown in Figure 3. Then we assign category labels to each image, including 9 super-classes and 179 sub-classes. Figure 2 (Right) illustrates the class breakdown of our CS12K dataset.

## 3.3 DATA ANALYSIS

For deeper insights into USCOD dataset, we compare our CS12K against 13 other related datasets including: (1) nine SOD datasets: SOD Movahedi & Elder (2010), PASCAL-S Li et al. (2014), ECSSD Yan et al. (2013), HKU-IS Li & Yu (2015), MSRA-B Liu et al. (2011), DUT-OMRON Yang et al. (2013), MSRA10K Cheng et al. (2015), DUTS Wang et al. (2017a), and SOC Fan et al. (2018a); (2) four COD datasets: CAMO Le et al. (2019), CHAMELEON Skurowski et al. (2018), COD10K Fan et al. (2020), and NC4K Lv et al. (2021); Table 1 shows the detailed information of these datasets. It can be seen that except for COD10K, all SOD datasets only contain salient objects, and all COD datasets only contain camouflaged objects. The scene of these datasets are relatively single. It is worth noting that, although the COD dataset COD10K contains some images with salient objects and images without any objects, these images lack labels and are not included in the training process. In contrast, the CS12K dataset we propose imposes no restrictions on scenes

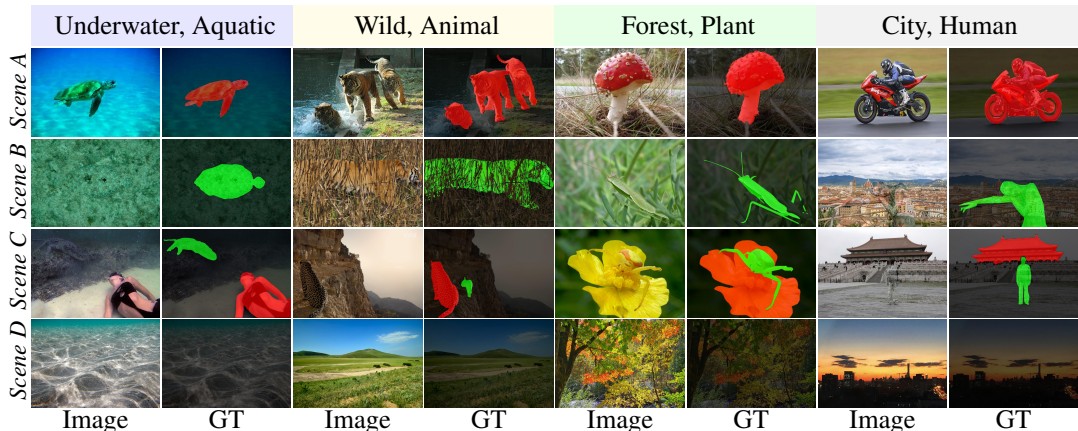

Figure 3: Example images from the CS12K dataset: Scene A: Only camouflaged object. Scene B: Only salient object. Scene C: Both salient and camouflaged objects simultaneously. Scene D: Background, with the absence of both types of objects. More examples can be found in Appendix.§E.

and includes labels for three attributes: saliency, camouflage, and background, with a well-balanced distribution. We aim to advance the field and explore effective methods for capturing camouflage and saliency patterns in unconstrained scenes.

## 4 PROPOSED USCNET BASELINE

**Overview.** As illustrated in Figure 4, the main components of the proposed USCNet include: (1) A SAM image encoder to extract object feature representation with adapter layers. (2) An Attribute-specific Prompt Generation (APG) that generates three discriminative prompts for each attribute: saliency, camouflage, and background. (3) A frozen mask decoder of SAM that is applied to predict the final saliency, camouflaged, and background masks based on different attribute prompts.

### 4.1 SEGMENT ANYTHING MODEL

SAM Kirillov et al. (2023) designs a flexible prompting-enabled model architecture for category-agnostic segmentation. Specifically, SAM consists of an image encoder, a prompt encoder, and a mask decoder. The image encoder is pre-trained using the Masked Auto Encoder (MAE) He et al. (2022), the prompt encoder handles dense and sparse inputs like boxes and points, and the mask decoder predicts the masks based on the encoded embeddings. In USCNet, we utilize the prompt architecture of SAM for identifying three attributes: saliency, camouflage, and background. The attribute prompts is generated by a designed APG module, eliminating the need for manual prompt. As a result, each attribute prompt is mapped to a distinct binary mask.

### 4.2 SAM ENCODER WITH ADAPTER

To leverage knowledge from SAM, SAM-Adapter Chen et al. (2023a) adapts SAM to downstream tasks and achieve enhanced performance with a parameter-efficient fine-tuning approach. Following that, USCNet integrates adapters into each layer of the SAM encoder, as depicted in Figure 4. As a result, the output image embedding $F$ from the tuned SAM Encoder exhibits features adept at addressing USCOD task. Through this approach, USCNet blending USCOD-specific knowledge with the general knowledge acquired by the larger model, better adapting to unconstrained scenes.

### 4.3 ATTRIBUTE-SPECIFIC PROMPT GENERATION

Our insight is that the detection of salient and camouflaged objects within a sample requires consideration of features across two dimensions: (i) Sample-generic features: For all samples, characteristics such as the size, position, color, and texture of the object serve as important generic features

Figure 4: Architecture of our USCNet. USCNet includes: SAM image encoder with adapter, Attribute-specific Prompt Generation (APG) module, and frozen SAM mask decoder.

for distinguishing salient and camouflaged objects. These features are applicable in most scenarios and can act as universal criteria for judgment; (ii) Sample-specific features: Relying solely on sample-generic features may not suffice in certain complex situations. For instance, when the salient and camouflaged objects share similar colors or categories, sample-generic features alone are insufficient for effective differentiation. In such cases, it is crucial to consider the specific contextual information within the sample and learn features that are closely associated with the current sample to assist in making an accurate judgment.

Based on this, we propose Attribute- specific Prompt Generation (APG) integrates both Dynamic Prompt Query (DPQ) and Static Prompt Query (SPQ) to generate discriminative attribute-specific prompts, where SPQ to extract sample-generic features, capturing attribute information that applies to all samples, and DPQ to extract sample-specific features, focusing on the unique contextual information of the current sample. Specifically, as depicted in Figure 4, the APG integrates both the Dynamic Prompt Query (DPQ) and Static Prompts Query (SPQ) to create attribute-specific prompts. The SPQ consists of a set of learnable query embeddings, which are designed to encapsulate general attributes. To formulate the DPQ, the system initially extracts features $F$ from the encoder to generate a coarse prediction, which is then processed through a sigmoid function to produce an attention map. This attention map is then element-wise multiplied with the original features $F$ to isolate attribute-specific features. These features are further refined through a linear layer to produce the DPQ, tailored to capture nuanced and specific attributes within individual samples. The DPQ generation process can be described by the formula:

$$[Q_{D\_S}, Q_{D\_C}, Q_{D\_B}] = MLP(\sigma(\Phi_{CH}(F)) \otimes F), \tag{1}$$

where $Q_{D\_S}$, $Q_{D\_C}$, and $Q_{D\_B}$ represent the DPQ for saliency, camouflage, and background, respectively. $MLP$ stands for a Multi-Layer Perceptron that processes the output. $\sigma$ denotes the sigmoid function, and $\Phi_{CH}$ represents the operation to predict a coarse prediction from the features $F$. The symbol $\otimes$ denotes element-wise multiplication. Unlike standard query embeddings which are fixed after training, the DPQ changes according to the sample, making it highly adaptable and capable of explicitly capturing the distinctive features of the camouflage and saliency across varying samples. The DPQ captures feature information from specific images, whereas the SPQ discerns the fundamental differences among three attributes. By combining the two, our APG attains improved performance. Subsequently, we employ self-attention to establish relationships between queries, and query-to-image ($Q2I$) attention to interact with image embedding, ultimately generating prompts for the three attributes: $P_S, P_C, P_B$. The process can be formulated as follows:

$$[P_S, P_C, P_B] = MLP(Q2I(SA(DPQ + SPQ), F)), \tag{2}$$

where $P_S$, $P_C$, and $P_B$ represent the prompts generated for identifying saliency, camouflage, and background elements, respectively. $SA$ represents the self-attention. $Q2I$ denotes the cross-attention from queries to the image embedding $F$, enabling the model to focus on relevant parts of the input based on the queries. Furthermore, we use a cross-attention from the image embedding to queries ($I2Q$) to focus on features related to attributes.

Based on the three attribute-specific prompts fed into the pre-trained mask decoder in SAM, three masks are obtained: $Mask\_S$, $Mask\_C$, and $Mask\_B$, representing the output saliency, camouflage, and background predictions, respectively. The process can be described as:

$$[Mask\_S, Mask\_C, Mask\_B] = MaskDe([P_S, P_C, P_B], F), \tag{3}$$

where MaskDe denotes frozen SAM mask decoder. Finally, a softmax function is applied to produce the final prediction.

Table 2: Quantitative comparisons with 19 related methods for USCOD. $\text{IoU}_S$ ↑: IoU score for salient objects. $\text{IoU}_C$ ↑: IoU score for camouflaged objects. The best two scores are highlighted in **red** and **green**, respectively. All metrics presented in the table are expressed as percentages (%). We use mIoU↑, mAcc↑, and CSCS↓ to evaluate the models in overall scenes.

| Task | Model | Venue | Update Para.(M) | Scene A $\text{IoU}_S$ | Scene B $\text{IoU}_C$ | Scene C $\text{IoU}_S$ | Scene C $\text{IoU}_C$ | Overall $\text{IoU}_S$ | Overall $\text{IoU}_C$ | mIoU | mAcc | CSCS |
|---|---|---|---|---|---|---|---|---|---|---|---|---|
| SOD | GateNet Zhao et al. (2020) | ECCV | 128 | 68.32 | 54.26 | 66.85 | 35.03 | 65.08 | 44.17 | 68.27 | 78.07 | 11.30 |
| | F3Net Wei et al. (2020) | AAAI | 26 | 70.05 | 52.62 | 67.20 | 36.38 | 66.12 | 44.81 | 68.80 | 77.86 | 9.36 |
| | MSFNet Zhang et al. (2021) | MM | 28 | 70.14 | 54.78 | 69.92 | 36.64 | 66.69 | 45.89 | 69.40 | 79.77 | 9.90 |
| | VST Liu et al. (2021a) | ICCV | 43 | 68.14 | 49.82 | 61.61 | 22.56 | 63.18 | 38.45 | 65.55 | 74.77 | 11.30 |
| | EDN Wu et al. (2022) | TIP | 43 | 71.59 | 57.94 | 69.37 | 37.70 | 68.00 | 48.27 | 70.70 | 80.60 | 9.23 |
| | ICON Zhuge et al. (2022) | TPAMI | 32 | 68.09 | 50.57 | 67.48 | 30.65 | 65.86 | 45.53 | 68.99 | 79.53 | 10.24 |
| COD | SINet-V2 Fan et al. (2021) | TPAMI | 27 | 72.96 | 56.16 | 67.21 | 36.06 | 69.50 | 47.47 | 70.20 | 79.58 | 8.83 |
| | PFNet Mei et al. (2021) | CVPR | 47 | 69.07 | 52.83 | 67.20 | 32.81 | 65.73 | 43.76 | 68.30 | 78.00 | 10.04 |
| | ZoomNet Pang et al. (2022) | CVPR | 33 | 74.11 | 51.12 | 66.79 | 29.69 | 66.43 | 43.28 | 68.35 | 77.72 | 8.88 |
| | FEDER He et al. (2023) | CVPR | 44 | 74.35 | 58.04 | 67.66 | 32.26 | 68.65 | 46.46 | 70.32 | 81.27 | 10.01 |
| | PRNet Hu et al. (2024) | TCSVT | 13 | 76.10 | 61.54 | 60.10 | 32.16 | 68.68 | 50.88 | 71.87 | 82.89 | 8.40 |
| | ICEG He et al. (2024) | ICLR | 100 | 73.67 | 68.38 | 68.43 | 44.33 | 69.22 | 58.71 | 74.68 | 83.53 | 8.16 |
| | CamoDiffusion Chen et al. (2023b) | AAAI | 72 | 75.01 | 59.39 | 53.49 | 45.03 | 63.49 | 52.80 | 70.70 | 77.73 | 7.73 |
| | CamoFormer Yin et al. (2024) | TPAMI | 71 | 75.88 | 66.19 | 73.33 | 44.14 | 71.86 | 56.09 | 74.81 | 84.17 | 7.57 |
| | PGT Wang et al. (2024) | CVIU | 68 | 72.75 | 61.51 | 70.01 | 41.21 | 71.46 | 56.83 | 75.03 | 83.35 | 9.09 |
| | SAM-Adapter Chen et al. (2023a) | ICCVW | 4.11 | 78.90 | 67.69 | 68.19 | 27.73 | 70.66 | 52.69 | 73.38 | 83.35 | 10.28 |
| | SAM2-Adapter Chen et al. (2024) | arXiv | 4.36 | 78.75 | 70.28 | 69.01 | 38.20 | 71.42 | 56.71 | 74.98 | 84.74 | 9.12 |
| Unified | EVP Liu et al. (2023) | CVPR | 4.95 | 75.85 | 59.81 | 71.41 | 37.64 | 70.30 | 50.36 | 72.16 | 79.96 | 8.67 |
| | VSCode Luo et al. (2024) | CVPR | 60 | 71.43 | 54.64 | 65.26 | 30.58 | 67.09 | 46.91 | 69.78 | 78.92 | 9.72 |
| USCOD | USCNet (Ours) | - | 4.04 | 79.70 | 74.99 | 74.80 | 45.73 | 75.57 | 61.34 | 78.03 | 87.92 | 7.49 |

## 4.4 LOSS FUNCTION

We use the ground truth (GT) to supervise the final prediction and coarse prediction. The total loss function of USCNet can be defined as:

$$L_{Total} = L_{CE}(I_{GT}, I_{Pred}) + L_{CE}(I_{GT}, I_{Coarse}), \tag{4}$$

where $I_{GT}, I_{Pred}$ and $I_{Coarse}$ respectively represent ground truth, final prediction, and the coarse prediction, while $L_{CE}$ represents the Cross Entropy loss.

## 5 CS12K BENCHMARK

As discussed above, our CS12K dataset is characterized by existence unconstrained, meaning each image may contain salient objects, camouflaged objects, both, or neither. Moreover, it covers categories spanning from salient to camouflaged objects. In CS12K benchmark, all models are trained and tested on the training set of CS12K (8,400 images) and the testing set of CS12K (3,600 images). To assess generalization, we also evaluated the model's performance across six widely used datasets. This includes common COD datasets such as COD10K, NC4K, and CAMO-TE, as well as popular SOD datasets like DUT-TE, HKU-IS, and DUT-OMRON. The results and specific settings of all generalization experiments are included in the Appendix.§C.

**Metrics.** Unlike the binary evaluation metrics widely used in SOD and COD (*e.g.* maximal F-measure Achanta et al. (2009)), the USCOD task involves three distinct attributes: saliency, camouflage, and background. To assess the performance of models tackling this multifaceted challenge, we leverage three established metrics for semantic segmentation: mean pixel accuracy of different categories (mAcc ↑), Intersection-over-Union of different categories (IoU ↑), and mean IoU (mIoU ↑). Inspired by Li et al. (2024), we also employ metrics AUC↑, SI-AUC↑, $F_m^{\beta}$↑, SI-$F_m^{\beta}$↑, $F_{\max}^{\beta}$↑, SI-$F_{\max}^{\beta}$↑, $E_m$↑ to evaluate the model's capability in detecting objects of varying sizes. Additionally, to evaluate the ability of the model to distinguish between salient and camouflaged objects, we propose a novel metric, the **C**amouflage-**S**aliency **C**onfusion **S**core (**CSCS** ↓), which is formulated as follows:

$$\text{CSCS} = \frac{1}{2}\left(\frac{\mathcal{P}_{CS}}{\mathcal{P}_{BS} + \mathcal{P}_{SS} + \mathcal{P}_{CS}} + \frac{\mathcal{P}_{SC}}{\mathcal{P}_{BC} + \mathcal{P}_{SC} + \mathcal{P}_{CC}}\right), \tag{5}$$

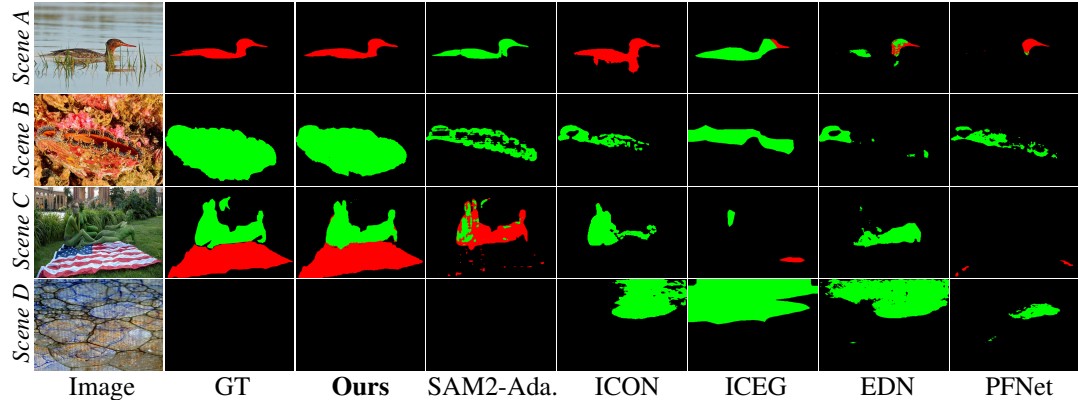

Figure 5: Qualitative comparisons of USCNet with five baselines across overall scenes. More visualization can be seen in Appendix.§F.

where $\mathbb{P} = \{\mathcal{P}_{\lambda\theta} \mid \lambda \in \Theta, \theta \in \Theta\}$, $\Theta = \{B, C, S\}$, the B, C and S denotes background, camouflage and saliency. As shown in Figure 6, $\mathcal{P}_{CS}$ represents regions where camouflage is predicted as salient, while $\mathcal{P}_{SC}$ represents regions where saliency is predicted as camouflage; both are regions of confusion. A lower CSCS indicates a stronger robustness to distinguish between salient and camouflaged objects. More details of CSCS can be seen in Appendix.§A.

**Competitors.** We compared our USCNet with 19 recent related models, including (I) SOD models: GateNet Zhao et al. (2020), F3Net Wei et al. (2020), MSFNet Zhang et al. (2021), VST Liu et al. (2021a), EDN Wu et al. (2022), ICON Zhuge et al. (2022); (II) COD models: SINet-V2 Fan et al. (2021), PFNet Mei et al. (2021), ZoomNet Pang et al. (2022),FEDER He et al. (2023), ICEG He et al. (2024), PRNet Hu et al. (2024), CamoDiffusion Chen et al. (2023b), CamoFormer Yin et al. (2024), PGT Wang et al. (2024), SAM-Adapter Chen et al. (2023a) and SAM2-Adapter Chen et al. (2024); (III) Unified methods: VSCode Luo et al. (2024) and EVP Liu et al. (2023).

**Technical Details.** All models are retrained using the training set of CS12K with an input image resolution of 352×352. Horizontal flipping and random cropping are applied for data augmentation. The experiments are conducted in PyTorch on one NVIDIA L40 GPU. The number of parameters fine-tuned for all models is detailed in Table 2. For our model, we use hiera-large version of SAM2 following the SAM2-Adapter Chen et al. (2024). AdamW optimizer is used a warm-up strategy and linear decay strategy. The initial learning rate is set to 0.0001. The batch size is set to 24, and the maximum number of epochs is set to 90. The technical details of all other comparison methods can be found in the Appendix.§D.

### 5.1 QUANTITATIVE EVALUATION

We present in Table 2 the performance of compared models on USCOD benchmark. Comparing the results of the models in single-attribute scenes (refer to Scene A and Scene B) with those in multi-attribute scenes (refer to Scene C) reveals that all models achieve lower scores in Scene C than in Scene A and Scene B. This indicates that the simultaneous presence of both salient objects and camouflaged objects increases the difficulty for the models to recognize both. Our method also achieves a greater lead in Scene C, *e.g.*, 74.80% on the $IoU_S$ and 45.73% on the $IoU_C$, demonstrating that our model is more adaptable when faced with more challenging scenarios. Furthermore, USC-Net achieves the best performance in all scenarios compared to all other compared methods. Additionally, the evaluation results for other metrics, including AUC↑, SI-AUC↑, $F_m^\beta$↑, SI-$F_m^\beta$↑, $F_{\max}^\beta$↑, SI-$F_{\max}^\beta$↑, and $E_m$↑, can be found in Table 4 and Table 5 in Appendix.§B.

### 5.2 QUALITATIVE EVALUATION

In Figure 5, we compare our qualitative results with SOD models (ICON Zhuge et al. (2022), EDN Wu et al. (2022)), COD model (ICEG He et al. (2024), PFNet Mei et al. (2021) and SAM2-Adapter Chen et al. (2024)). In the Scene A and Scene B of Figure 5, our method exhibited a better

Table 3: **Left:** Performance of different base model. *In the original SAM or SAM2, we only fine-tune the mask decoder. **Right:** Effectiveness of different components in APG. DPQ: dynamic prompt query. SPQ: static prompt query. Q2I: query-to-image attention. I2Q: image-to-query attention. Para.: update parameter (M). All metrics tested on the overall scenes test set and presented in the table are expressed as percentages (%).

| Method | Base | Para. | $IoU_S$ | $IoU_C$ | mIoU | mAcc | CSCS |
|---|---|---|---|---|---|---|---|
| SAM* | SAM | 3.92 | 51.07 | 33.00 | 59.56 | 68.73 | 18.66 |
| USCNet | SAM | 4.08 | 73.93 | 56.50 | 75.87 | 83.86 | 8.24 |
| SAM2* | SAM2 | 4.22 | 66.42 | 44.02 | 68.78 | 77.65 | 11.58 |
| **USCNet** | SAM2 | 4.04 | **75.57** | **61.34** | **78.03** | **87.92** | **7.49** |

| Encoder | Decoder | DPQ | SPQ | Q2I | I2Q | Para. | $IoU_S$ | $IoU_C$ | mIoU | mAcc | CSCS |
|---|---|---|---|---|---|---|---|---|---|---|---|
| Frozen | Tuning | ✗ | ✗ | ✗ | ✗ | 4.22 | 66.42 | 44.02 | 68.78 | 77.65 | 11.58 |
| Tuning | Tuning | ✗ | ✗ | ✗ | ✗ | 4.36 | 71.42 | 56.71 | 74.98 | 84.74 | 9.12 |
| Tuning | Frozen | ✗ | ✓ | ✓ | ✓ | 3.44 | 71.68 | 57.53 | 75.31 | 85.15 | 9.07 |
| Tuning | Frozen | ✓ | ✗ | ✓ | ✓ | 4.03 | 74.32 | 58.91 | 76.96 | 85.80 | 7.98 |
| Tuning | Frozen | ✓ | ✓ | ✗ | ✗ | 0.75 | 70.97 | 56.56 | 74.77 | 84.43 | 9.85 |
| Tuning | Frozen | ✓ | ✓ | ✓ | ✗ | 2.40 | 73.08 | 58.45 | 76.73 | 85.63 | 8.52 |
| Tuning | Frozen | ✓ | ✓ | ✓ | ✓ | 4.04 | **75.57** | **61.34** | **78.03** | **87.92** | **7.49** |

detection capability for salient objects or camouflaged objects. Benefiting from the APG module, our method better distinguished salient objects and camouflaged objects in the same image within Scene C of Figure 5. For Scene D, SOD and COD methods become confused when encountering backgrounds, resulting in poor performance and unstable robustness, whereas our model demonstrates better performance in this scenario. More qualitative evaluation can be seen in Appendix.§F.

## 5.3 ABLATION STUDY

**Performance of Different Base Models.** We conducted ablation experiments to evaluate the performance of different base models, as presented in Table 3.(**Left**). First, as shown in the first two and last two rows of the table, our model demonstrates significant performance improvements on the USCOD benchmark, regardless of whether SAM Kirillov et al. (2023) (default vit-huge version) or SAM2 Ravi et al. (2024) (default hiera-large version) is used as the base model. For instance, when using SAM as the base model, our method achieves a 16.31% gain in mIoU compared to the original SAM, while utilizing SAM2 results in a 9.25% improvement in mIoU over the original SAM2. Additionally, transitioning from SAM to SAM2 (as shown in rows 2 and 4) results in performance gains across all metrics with fewer fine-tuned parameters.

**Effectiveness of Different Components in APG.** As we shown in Table 3. (**Right**), ablation experiments were conducted to validate the effectiveness of the proposed components in APG module. From the third, fourth, and seventh rows, it is evident that both DPQ and SPQ improve the performance of model, with DPQ providing a greater performance enhancement than SPQ when used together, achieving optimal results. The fifth, sixth, and seventh rows demonstrate that Q2I and I2Q also facilitate the distinction of salient camouflaged objects, leading to reductions in CSCS of 2.36% and 1.08%, respectively. Additionally, compared to the original SAM2 (refer to line 1) and SAM2-Adapter (refer to line 2), the proposed USCNet enhances performance on the USCOD task across all metrics through a more efficient fine-tuning approach by incorporating the APG module and freezing the mask decoder.

## 6 CONCLUSION

We analyze and address the limitations of classical SOD and COD tasks, which restrict research to scenarios with only salient or only camouflaged objects. Based on that, a new benchmark called Unconstrained Salient and Camouflaged Object Detection (USCOD), is defined to allow for the unrestricted presence of both salient and camouflaged objects within images. We propose a new evaluation metric, *i.e.*, the Camouflage-Saliency Confusion Score (CSCS), to assess the confusion of the model between camouflaged and salient objects. To support research on USCOD, we have constructed a large-scale dataset, CS12K, that features a diverse range of scenes and categories. We introduce a baseline method, USCNet, which decouples mask reconstruction from attribute distinction to focus on learning the differences between saliency and camouflage patterns, achieving state-of-the-art performance on the USCOD task. The proposed USCOD reduces reliance on specific scenarios, increasing the representation of scenes where both salient and camouflaged objects coexist, as well as scenes where neither is present, thus enhancing generalizability across diverse natural environments.

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

APPENDIX

**Table of contents:**

## A   CSCS METRIC

Contrary to the Intersection over Union (IoU) that measures accuracy for a single class, the Camouflage-Saliency Confusion Score (CSCS) assesses the misclassification between two distinct classes. The CSCS, designed to evaluate the confusion between camouflaged and salient objects, is calculated as follows:

$$\text{CSCS} = \frac{1}{2}\left(\frac{\mathcal{P}_{CS}}{\mathcal{P}_{BS} + \mathcal{P}_{SS} + \mathcal{P}_{CS}} + \frac{\mathcal{P}_{SC}}{\mathcal{P}_{BC} + \mathcal{P}_{SC} + \mathcal{P}_{CC}}\right), \tag{6}$$

where $\mathbb{P} = \{\mathcal{P}_{\lambda\theta} \mid \lambda \in \Theta, \theta \in \Theta\}$, $\Theta = \{B, C, S\}$, the B, C and S denote background, camouflage and saliency. A lower CSCS value indicates a stronger ability of the network to discriminate between salient and camouflaged objects. $\mathcal{P}_{CS}$ represents the label as camouflage but is predicted as saliency. We aim to minimize the misclassification of camouflaged pixels as salient, ensuring the network correctly distinguishes between camouflaged and salient objects. The same applies to $\mathcal{P}_{SC}$. As shown in Figure 7, we present the confusion matrix of the proposed USCNet on the CS12K test set. Our model balances improvements across all metrics, achieving a mIoU of 0.775 and a CSCS of 0.0749 (see Table 2 in the manuscript).

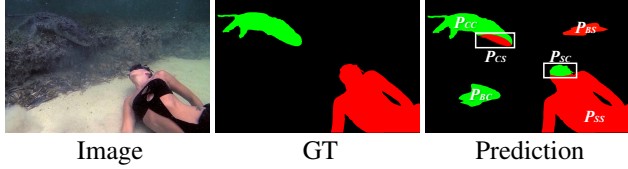

Image        GT        Prediction

Figure 6: The illustration of $\mathcal{P}_{BS}$, $\mathcal{P}_{SS}$, $\mathcal{P}_{CS}$, $\mathcal{P}_{BC}$, $\mathcal{P}_{SC}$, and $\mathcal{P}_{CC}$ in the CSCS metric. The **red** mask represents the salient regions, and the **green** mask denotes the camouflaged regions.

## B   PERFORMANCE OF MODELS IN DETECTING OBJECTS OF VARYING SIZES

To evaluate the model's ability to detect objects of varying sizes, we employ several metrics: AUC↑, SI-AUC↑, $F_m^\beta$↑, SI-$F_m^\beta$↑, $F_{\max}^\beta$↑, SI-$F_{\max}^\beta$↑, $E_m$↑. From Table 4 and Table 5, it can be observed that, compared to the size-sensitive(e.g., AUC↑ and $F_m^\beta$↑) and size-invariance metrics(e.g., SI-AUC↑ and SI-$F_m^\beta$↑), our method exhibits smaller performance fluctuations, demonstrating its robustness to variations in object size and number in the scene.

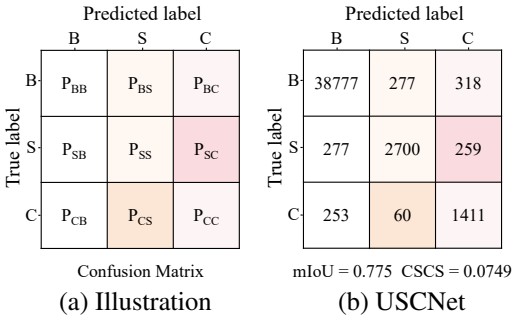

Figure 7: Confusion matrix of our USCNet on the CS12K test set. The units of the values in the confusion matrix are in tens of thousands (**1E+04**).

Table 4: Performance of different models detecting salient objects on CS12K testing set.

| Task | Model | Update Params(M) | CS12K-SOD | | | | | | |
|------|-------|:-:|:-:|:-:|:-:|:-:|:-:|:-:|:-:|
| | | | AUC↑ | SI-AUC↑ | $F_m^\beta$↑ | SI-$F_m^\beta$↑ | $F_{max}^\beta$↑ | SI-$F_{max}^\beta$↑ | $E_m$↑ |
| SOD | GateNet Zhao et al. (2020) | 128 | .810 | .812 | .696 | .754 | .706 | .764 | .775 |
| | F3Net Wei et al. (2020) | 26 | .828 | .826 | .722 | .765 | .734 | .777 | .803 |
| | MSFNet Zhang et al. (2021) | 28 | .832 | .831 | .726 | .772 | .735 | .782 | .805 |
| | VST Liu et al. (2021a) | 43 | .777 | .777 | .642 | .732 | .650 | .741 | .742 |
| | EDN Wu et al. (2022) | 43 | .831 | .830 | .726 | .769 | .736 | .780 | .804 |
| | ICON Zhuge et al. (2022) | 32 | .821 | .832 | .702 | .764 | .711 | .774 | .795 |
| COD | SINetV2 Fan et al. (2021) | 27 | .843 | .842 | .755 | .783 | .765 | .793 | .827 |
| | PFNet Mei et al. (2021) | 47 | .820 | .822 | .712 | .756 | .724 | .767 | .799 |
| | ZoomNet Pang et al. (2022) | 33 | .821 | .823 | .710 | .765 | .720 | .774 | .791 |
| | FEDER He et al. (2023) | 44 | .841 | .842 | .742 | .784 | .750 | .796 | .820 |
| | ICEG He et al. (2024) | 100 | .830 | .825 | 734 | .762 | .743 | .770 | .831 |
| | PRNet Hu et al. (2024) | 13 | .851 | .845 | .742 | .779 | .750 | .792 | .832 |
| | CamoFormer Yin et al. (2024) | 71 | .844 | .843 | .750 | .782 | .758 | .790 | .821 |
| | PGT Wang et al. (2024) | 68 | .831 | .828 | .717 | .773 | .727 | .784 | .791 |
| | SAM2-Adapter Chen et al. (2024) | 4.36 | .847 | .847 | .741 | .783 | .751 | .794 | .816 |
| Unified | VSCode Luo et al. (2024) | 60 | .826 | .830 | .720 | .769 | .731 | .790 | .802 |
| | EVP Liu et al. (2023) | 4.95 | .850 | .847 | .751 | .782 | .771 | .792 | .830 |
| USCOD | USCNet(ours) | **4.04** | **.853** | **.850** | **.761** | **.787** | **.772** | **.798** | **.833** |

Table 5: Performance of different models detecting camouflaged objects on CS12K testing set.

| Task | Model | Update Params(M) | CS12K-COD | | | | | | |
|------|-------|:-:|:-:|:-:|:-:|:-:|:-:|:-:|:-:|
| | | | AUC↑ | SI-AUC↑ | $F_m^\beta$↑ | SI-$F_m^\beta$↑ | $F_{max}^\beta$↑ | SI-$F_{max}^\beta$↑ | $E_m$↑ |
| SOD | GateNet Zhao et al. (2020) | 128 | .692 | .687 | .443 | .558 | .453 | .569 | .651 |
| | F3Net Wei et al. (2020) | 26 | .695 | .687 | .449 | .564 | .458 | .574 | .649 |
| | MSFNet Zhang et al. (2021) | 28 | .698 | .691 | .455 | .565 | .465 | .576 | .659 |
| | VST Liu et al. (2021a) | 43 | .626 | .625 | .303 | .536 | .312 | .546 | .524 |
| | EDN Wu et al. (2022) | 43 | .709 | .703 | .476 | .575 | .485 | .585 | .670 |
| | ICON Zhuge et al. (2022) | 32 | .663 | .663 | .384 | .549 | .394 | .560 | .587 |
| COD | SINetV2 Fan et al. (2021) | 27 | .715 | .705 | .505 | .588 | .514 | .598 | .690 |
| | PFNet Mei et al. (2021) | 47 | .678 | .672 | .429 | .544 | .440 | .555 | .630 |
| | ZoomNet Pang et al. (2022) | 33 | .657 | .653 | .394 | .545 | .405 | .556 | .588 |
| | FEDER He et al. (2023) | 44 | .710 | .703 | .486 | .567 | .497 | .578 | .689 |
| | ICEG He et al. (2024) | 100 | .730 | .717 | .525 | .601 | .532 | .609 | .719 |
| | PRNet Hu et al. (2024) | 13 | .705 | .695 | .454 | .569 | .464 | .579 | .652 |
| | CamoFormer Yin et al. (2024) | 71 | .756 | .745 | .565 | .626 | .575 | .636 | .743 |
| | PGT Wang et al. (2024) | 68 | .746 | .734 | .527 | .596 | .539 | .607 | .715 |
| | SAM2-Adapter Chen et al. (2024) | 4.36 | .770 | .761 | .575 | .637 | .585 | .647 | .746 |
| Unified | VSCode Luo et al. (2024) | 60 | .735 | .727 | .519 | .601 | .525 | .597 | .722 |
| | EVP Liu et al. (2023) | 4.95 | .695 | .684 | .485 | .577 | .494 | .587 | .650 |
| USCOD | USCNet(ours) | **4.04** | **.801** | **.794** | **.610** | **.658** | **.619** | **.667** | **.795** |

## C  RESULTS ON COD AND SOD DATASETS

To further validate the effectiveness and robustness of our method regarding generalizability, we conduct tests on popular SOD datasets (DUTS Wang et al. (2017a), HKU-IS Li & Yu (2015), and

Table 6: Quantitative comparisons with related methods on the DUTS, HKU-IS, and DUT-OMRON test sets. ↑ / ↓ represents the higher/lower the score, the better.

| Task | Model | Update Params(M) | DUTS $F_\beta^{max}\uparrow$ | $F_\beta^\omega\uparrow$ | $M\downarrow$ | $S_\alpha\uparrow$ | $E_\phi^m\uparrow$ | HKU-IS $F_\beta^{max}\uparrow$ | $F_\beta^\omega\uparrow$ | $M\downarrow$ | $S_\alpha\uparrow$ | $E_\phi^m\uparrow$ | DUT-OMRON $F_\beta^{max}\uparrow$ | $F_\beta^\omega\uparrow$ | $M\downarrow$ | $S_\alpha\uparrow$ | $E_\phi^m\uparrow$ |
|---|---|---|---|---|---|---|---|---|---|---|---|---|---|---|---|---|---|
| SOD | GateNet Zhao et al. (2020) | 128 | .666 | .644 | .062 | .755 | .765 | .804 | .785 | .049 | .841 | .857 | .634 | .603 | .079 | .747 | .751 |
| | F3Net Wei et al. (2020) | 26 | .703 | .683 | .055 | .783 | .794 | .832 | .816 | .044 | .853 | .881 | .638 | .615 | .073 | .747 | .758 |
| | MSFNet Zhang et al. (2021) | 28 | .651 | .638 | .063 | .749 | .758 | .824 | .806 | .045 | .853 | .877 | .641 | .611 | .076 | .751 | .764 |
| | VST Liu et al. (2021a) | 43 | .630 | .610 | .061 | .744 | .749 | .777 | .760 | .052 | .820 | .851 | .580 | .560 | .073 | .720 | .715 |
| | EDN Wu et al. (2022) | 43 | .692 | .676 | .053 | .784 | .785 | .820 | .806 | .043 | .852 | .873 | .616 | .597 | .071 | .742 | .735 |
| | ICON Zhuge et al. (2022) | 32 | .679 | .647 | .069 | .769 | .785 | .814 | .787 | .051 | .843 | .874 | .615 | .576 | .099 | .728 | .738 |
| COD | SINetV2 Fan et al. (2021) | 27 | .732 | .710 | .052 | .801 | .821 | .838 | .822 | .046 | .847 | .884 | .665 | .642 | .068 | .763 | .786 |
| | PFNet Mei et al. (2021) | 47 | .691 | .668 | .060 | .775 | .790 | .818 | .801 | .048 | .843 | .876 | .643 | .614 | .075 | .747 | .764 |
| | ZoomNet Pang et al. (2022) | 33 | .729 | .709 | .053 | .801 | .813 | .785 | .774 | .051 | .830 | .842 | .623 | .601 | .075 | .742 | .735 |
| | FEDER He et al. (2023) | 44 | .736 | .714 | .052 | .808 | .821 | .839 | .827 | .045 | .869 | .881 | .645 | .615 | .077 | .755 | .760 |
| | PRNet Hu et al. (2024) | 13 | .773 | .756 | .043 | .830 | .849 | .840 | .833 | .044 | .857 | .880 | .708 | .685 | .057 | **.796** | .808 |
| | ICEG He et al. (2024) | 100 | .719 | .700 | .050 | .789 | .820 | .832 | .815 | .045 | .848 | **.896** | .664 | .645 | .061 | .762 | .785 |
| | CamoFormer Yin et al. (2024) | 71 | .733 | .715 | .049 | .813 | .819 | .838 | .817 | .046 | .857 | .884 | .687 | .661 | .066 | .783 | .793 |
| | PGT Wang et al. (2024) | 68 | .686 | .670 | .053 | .786 | .779 | .819 | .802 | .044 | .855 | .871 | .642 | .619 | .068 | .758 | .754 |
| | SAM-Adapter Chen et al. (2023a) | 4.11 | .761 | .746 | .048 | .834 | .796 | .822 | .806 | .043 | .836 | .869 | .708 | .685 | .059 | .793 | .802 |
| | SAM2-Adapter Chen et al. (2024) | 4.36 | .776 | .762 | .041 | .831 | .848 | .831 | .828 | **.042** | .849 | .881 | .706 | .692 | **.056** | .790 | .810 |
| Unified | VSCode Luo et al. (2024) | 60 | .724 | .706 | .060 | .795 | .812 | .834 | .830 | .043 | .851 | .885 | .636 | .608 | .075 | .748 | .753 |
| | EVP Liu et al. (2023) | 4.95 | .769 | .750 | .045 | .833 | .836 | .835 | .832 | .043 | .852 | .878 | **.710** | .692 | .057 | .794 | .810 |
| USCOD | USCNet(ours) | **4.04** | **.784** | **.780** | **.040** | **.835** | **.852** | **.844** | **.840** | **.042** | **.860** | .886 | **.710** | **.697** | **.056** | **.796** | **.814** |

Table 7: Quantitative comparisons with ten related methods on CAMO , COD10K , and NC4K test set. ↑ / ↓ represents the higher/lower the score, the better.

| Task | Model | Update Params(M) | CAMO $F_\beta^{max}\uparrow$ | $F_\beta^\omega\uparrow$ | $M\downarrow$ | $S_\alpha\uparrow$ | $E_\phi^m\uparrow$ | NC4K $F_\beta^{max}\uparrow$ | $F_\beta^\omega\uparrow$ | $M\downarrow$ | $S_\alpha\uparrow$ | $E_\phi^m\uparrow$ | COD10K $F_\beta^{max}\uparrow$ | $F_\beta^\omega\uparrow$ | $M\downarrow$ | $S_\alpha\uparrow$ | $E_\phi^m\uparrow$ |
|---|---|---|---|---|---|---|---|---|---|---|---|---|---|---|---|---|---|
| SOD | GateNet Zhao et al. (2020) | 128 | .573 | .542 | .109 | .666 | .680 | .562 | .529 | .047 | .707 | .724 | .675 | .645 | .066 | .752 | .777 |
| | F3Net Wei et al. (2020) | 26 | .538 | .506 | .117 | .643 | .657 | .576 | .539 | .047 | .712 | .744 | .661 | .633 | .070 | .738 | .773 |
| | MSFNet Zhang et al. (2021) | 28 | .568 | .535 | .113 | .661 | .682 | .543 | .534 | .052 | .692 | .719 | .671 | .645 | .067 | .747 | .778 |
| | VST Liu et al. (2021a) | 43 | .484 | .455 | .109 | .661 | .631 | .468 | .430 | .055 | .661 | .670 | .597 | .567 | .072 | .710 | .732 |
| | EDN Wu et al. (2022) | 43 | .573 | .542 | .109 | .666 | .680 | .595 | .562 | .044 | .727 | .756 | .688 | .660 | .063 | .761 | .795 |
| | ICON Zhuge et al. (2022) | 32 | .520 | .481 | .125 | .641 | .648 | .540 | .502 | .053 | .695 | .715 | .631 | .596 | .076 | .724 | .752 |
| COD | SINetV2 Fan et al. (2021) | 27 | .590 | .562 | .102 | .681 | .694 | .609 | .577 | .043 | .729 | .763 | .662 | .639 | .066 | .740 | .769 |
| | PFNet Mei et al. (2021) | 47 | .535 | .505 | .110 | .652 | .661 | .556 | .524 | .049 | .699 | .730 | .660 | .633 | .068 | .737 | .769 |
| | ZoomNet Pang et al. (2022) | 33 | .494 | .472 | .113 | .635 | .612 | .520 | .496 | .048 | .488 | .671 | .596 | .576 | .074 | .708 | .706 |
| | FEDER He et al. (2023) | 44 | .567 | .538 | .106 | .669 | .687 | .636 | .598 | .042 | .749 | .793 | .688 | .664 | .063 | .758 | .790 |
| | PRNet Hu et al. (2024) | 13 | .648 | .607 | .096 | .716 | .766 | .709 | .672 | .059 | .772 | .820 | .650 | .603 | .038 | .756 | .815 |
| | ICEG He et al. (2024) | 100 | .728 | .697 | .066 | .769 | .820 | .735 | .708 | .051 | .786 | .840 | .645 | .610 | .035 | .753 | .807 |
| | CamoFormer Yin et al. (2024) | 71 | .645 | .618 | .078 | .732 | .750 | .729 | .707 | .054 | .789 | .822 | .668 | .639 | .035 | .770 | .811 |
| | PGT Wang et al. (2024) | 68 | .635 | .612 | .089 | .718 | .730 | .729 | .706 | .052 | .791 | .819 | .642 | .612 | .036 | .758 | .786 |
| | SAM-Adapter Chen et al. (2023a) | 4.11 | .661 | .638 | .080 | .744 | .753 | .688 | .667 | .037 | .788 | .808 | .727 | .710 | .051 | .794 | .809 |
| | SAM2-Adapter Chen et al. (2024) | 4.36 | .717 | .692 | .074 | .779 | .807 | .724 | .694 | .044 | .809 | .847 | .735 | .694 | .045 | .819 | .845 |
| Unified | VSCode Luo et al. (2024) | 60 | .562 | .532 | .109 | .658 | .678 | .626 | .591 | .043 | .744 | .787 | .684 | .662 | .067 | .753 | .783 |
| | EVP Liu et al. (2023) | 4.95 | .636 | .637 | .085 | .701 | .718 | .693 | .694 | .040 | .742 | .775 | .615 | .614 | .069 | .724 | .749 |
| USCOD | USCNet(ours) | **4.04** | **.829** | **.790** | **.049** | **.845** | **.886** | **.794** | **.768** | **.039** | **.839** | **.877** | **.743** | **.700** | **.030** | **.821** | **.869** |

DUT-OMRON Yang et al. (2013)) and COD datasets (CAMO Le et al. (2019), COD10K Fan et al. (2020), and NC4K Lv et al. (2021)), with all methods uniformly trained using our CS12k dataset. We adopt five metrics that are widely used in COD and SOD tasks Wang et al. (2021); Fan et al. (2021). These metrics include maximal F-measure ($F_\beta^{max}$ ↑) Achanta et al. (2009), weighted F-measure ($F_\beta^\omega$ ↑) Margolin et al. (2014), Mean Absolute Error (MAE, $M$ ↓) Perazzi et al. (2012), Structural measure (S-measure, $S_\alpha$ ↑) Fan et al. (2017), and mean Enhanced alignment measure (E-measure, $E_\phi^m$ ↑) Fan et al. (2018b). As shown in Table 6 and Table 7, our USCNet achieves state-of-the-art performance on these datasets through parameter-efficient fine-tuning. This further confirms the strong capability of our method to accurately identify both salient and camouflaged objects in unconstrained environments. This achievement is attributed to the exceptional versatility of SAM in class-agnostic segmentation tasks and the discriminative ability of our specially designed APG for distinguishing between salient and camouflaged objects.

# D    MORE TECHNICAL DETAILS

**Backbone of models.** The models compared can be divided into two categories based on their papers: one is full-tuning models, and the other is parameter-efficient fine-tuning (PEFT) models.(i)Full Tuning models: Include all SOD and COD methods and VSCode in the Unified Method. For fairness, the models compared are all trained according to the configurations specified in their original papers. (ii)PEFT models: SAM-Adapter, SAM2-Adapter, EVP in the Unified Method and our model. The backbone architectures across various models consist of several types. For full tuning, VST employs a transformer encoder based on T2T-ViT Yuan et al. (2021), while SINet-V2utilizes Res2Net-50 Gao et al. (2019). VSCode uses Swin-T Liu et al. (2021b), and ICEG adopts Swin-B Liu et al. (2021b). PRNet is based on the SMT backbone Lin et al. (2023), and both CamoD-iffusion, CamoFormer, and PGT use PVTv2-b4 Wang et al. (2022). Other models generally rely on ResNet-50 He et al. (2016) with pre-trained weights from ImageNet Deng et al. (2009). In the case of PEFT models, EVP uses SegFormer-B4 Xie et al. (2021) as its base, SAM-Adapter uses the default ViT-H version of SAM Kirillov et al. (2023), and both SAM2-Adapter and our model employ the hiera-large version of SAM2 Ravi et al. (2024).

**Training and Inference.** For traditional SOD and COD models: The task of USCOD is defined by three attributes: saliency, camouflage, and background. Conventional methods for COD and SOD are crafted for dichotomous mapping tasks and don't seamlessly transition to the nuanced demands of USCOD. Inspired by seminal works in semantic segmentations Long et al. (2015); Strudel et al. (2021), we retool the output layers of our models to yield a tripartite representation for saliency, camouflage, and background. This is achieved by harnessing a softmax layer to generate a predictive mapping. We employ a cross-entropy loss function to refine the model, which is congruent with our overarching methodological framework. For unified models: VSCode and EVP, which require task-specific prompts for each dataset, we create two copies of the CS12K training set. One copy is used for SOD, with the ground truth being the SOD-only mask, and is used to train the prompts corresponding to the SOD task. The other copy is used for COD, with the ground truth being the COD-only mask, and is used to train the prompts corresponding to the COD task.VSCode is trained once using all 16,800 images (two copies of 8,400 images), while EVP is trained twice on the two separate training sets (each containing 8,400 images) to obtain the two task-specific prompts. During Inference, all unified models perform inference on the testing set of CS12K twice, with the corresponding prompt enabled for each task. The first inference run generates the SOD results, and the second inference run generates the COD results. The final prediction is obtained by merging the SOD and COD predictions. For overlapping pixels, the attribute with the higher prediction value between the two tasks is chosen as the final attribute for that pixel.

# E    MORE CS12K DATASET DETAIL AND EXAMPLES

**Object number distribution.** Our CS12K dataset contains images with different numbers of objects. To show it more clearly, we have counted the distribution of images with different numbers of objects in CS12K, as shown in the following Table 8.

Table 8: Distribution of Images with Different Numbers of Objects in CS12K.

| Number of objects | 0 | 1 | 2 | >2 |
|---|---|---|---|---|
| Number of images | 3000 | 4197 | 2335 | 2468 |

**Detail of annotation process.** For Scene A and B, we retained their original annotations, while Scene D did not require additional annotation. Therefore, we focus here on detailing the annotation process for Scene C.

- **Initial Determination of Object Attributes:** We invited 7 observers to perform the initial identification of salient and camouflaged objects in the images. A voting process was used to determine the salient and camouflaged objects in each image, with objects and their attributes receiving more than half of the votes being retained. We then used Photoshop to apply red boxes for salient objects and green boxes for camouflaged objects, which served as the reference for the subsequent mask annotation step.

- **Mask Annotation:** We invited 9 volunteers to perform detailed mask annotation for the dataset using the ISAT interactive annotation tool Ji & Zhang (2023), which supports SAM semi-automatic labeling.
- **Annotation Quality Control:** After annotation, we invited an additional 3 observers to review and refine the results. Masks with imprecise or incorrect annotations were manually corrected.

**More CS12K examples.** In Figure 8, we illustrate a selection of images from the CS12K dataset, each featuring both salient and camouflaged objects. The main difference between our CS12K dataset and existing SOD and COD datasets is that it includes a curated subset of 3,000 images, each featuring both salient and camouflaged objects. We invest significant time and effort in finding and annotating these images. Our dataset spans an extensive variety of environments, including, but not limited to, terrestrial, aquatic, alpine, sylvan, and urban ecosystems, and encompasses a broad spectrum of categories, such as lion, flower and various fruit species. This dataset is designed to assist the SOD and COD research communities in advancing the state-of-the-art in discerning more sophisticated saliency and camouflage patterns.

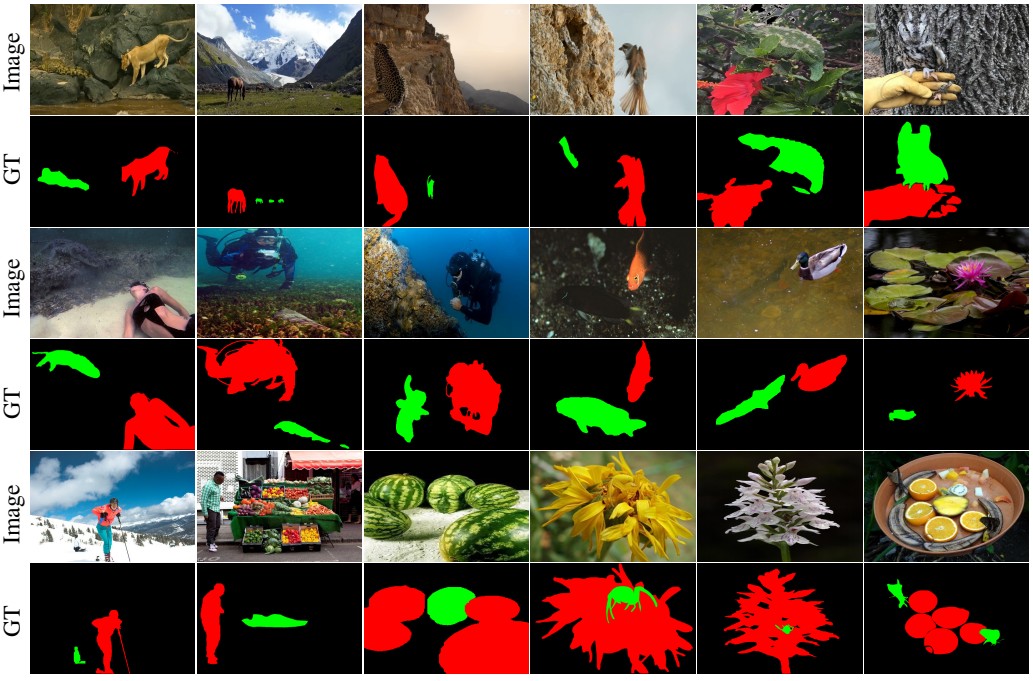

Figure 8: Additional Example images where exist both camouflaged and salient objects from the CS12K dataset. Our collection comprises 3,000 carefully curated and annotated images, encompassing a diverse range of scenes and categories. **Please zoom in for an enhanced view**.

## F ADDITIONAL QUALITATIVE RESULTS

We present additional predictive results of our USCNet model compared to other COD and SOD models in the CS12K test set. As illustrated in Figure 11, our model outperforms its competitors. Specifically, across four different scenes, our model demonstrates a high degree of consistency with the ground truth, especially in distinguishing between salient and camouflaged objects. Our model is adept at learning distinctive features of saliency and camouflage. For instance, it can accurately identify patterns such as camouflaged humans (refer to the fifth column of Figure 11). Moreover, in scenes devoid of salient or camouflaged objects, our model remains unaffected by complex backgrounds (refer to the sixth column of Figure 11). This further underscores the robustness and accuracy of our USCNet model.

## G PRACTICAL APPLICATIONS OF USCOD

**Military Surveillance and Enemy Reconnaissance.** In a military environment, salient objects might include large military equipment such as vehicles, tanks, helicopters, etc., while camouflaged objects could be soldiers or equipment hidden in vegetation or camouflage materials. The simultaneous detection of both salient and camouflaged objects helps enhance battlefield situational awareness and prevents overlooking potential threats.

**Post-Disaster Search and Rescue.** After a disaster, salient objects might include obvious signs of life in rubble (such as clearly visible trapped individuals), while camouflaged objects could be life signs that are difficult to detect due to obstruction or chaotic environments (such as partially buried survivors). The simultaneous detection of both salient and camouflaged objects is crucial for improving search and rescue efficiency.

**Ecological Protection and Wildlife Monitoring.** In natural environments, salient objects might be easily visible animals (such as birds in open areas), while camouflaged objects could be animals hidden in vegetation (such as insects with protective coloration). The simultaneous detection of both salient and camouflaged objects allows for more comprehensive wildlife population surveys and ecological research.

**Multi-Level Lesion Detection.** In medical imaging, detecting both salient lesions (such as obvious tumors or organ damage) and camouflaged lesions (such as those blurred by background textures or early-stage lesions) helps doctors more thoroughly assess a patient's health condition.

**Diving Hazard Warnings.** During diving, salient objects might include coral or schools of fish that attract the diver's attention, while camouflaged objects could be hidden dangerous creatures (such as stonefish or moray eels). The simultaneous detection of both salient and camouflaged objects helps guide divers in more comprehensively avoiding dangers.

## H DIFFICULTY OF USCOD

As illustrated in Figure 11, most methods encounter difficulties in distinguishing salient objects from camouflaged on CS12K benchmark. The essence of the challenge in USCOD lies in differentiating between salient and camouflaged objects within unconstrained scenes, mirroring the capabilities of human vision. Moving beyond the simplicity of traditional classification tasks, distinguishing between visual saliency and camouflage requires a deeper semantic insight. Our observation indicates that when a vanilla network architecture is used for the USCOD task, a decoder responsible for differentiating, localizing, and segmenting both salient and camouflaged objects encounters challenges in acquiring highly discriminative visual features. To address this, our approach leverages a specialized decoder focused on precise localization and segmentation. This allows for more effective learning of the subtle distinctions between saliency and camouflage patterns, enhancing the ability to discern and differentiate these complex visual cues. Figure 9 showcases our architectural innovation, incorporating a frozen, pre-trained SAM mask decoder and an APG representing our venture into mining highly discriminative features. This design separates feature analysis from object segmentation, enabling our model to focus on and extract distinct attributes crucial for differentiating saliency from camouflage patterns, thereby improving its performance in complex visual environments.

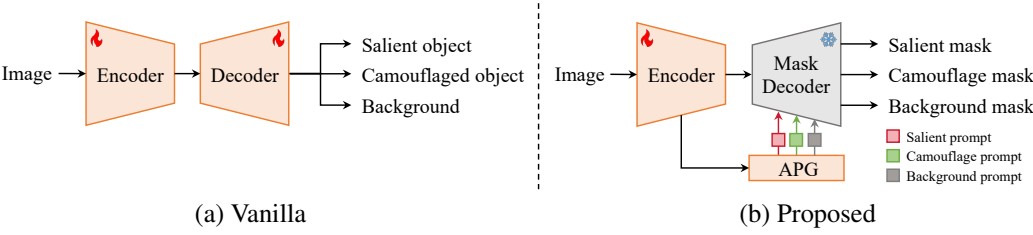

(a) Vanilla        (b) Proposed

Figure 9: (a) In Vanilla network architecture, a decoder is used for both localization and segmentation. (b) In our proposed architecture, the APG is used for localizing salient and camouflaged objects, while the mask decoder is used for segmentation.

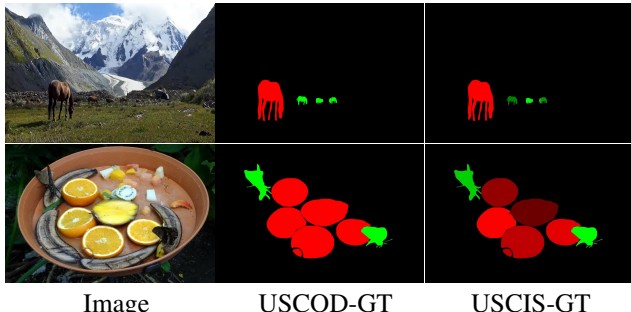

Image                     USCOD-GT                     USCIS-GT

Figure 10: An illustration of the USCOD and USCIS tasks: In contrast to USCOD, USCIS requires not only the identification of salient versus camouflaged objects but also the discrimination between individual instances of saliency and camouflage.

# I LIMITATIONS

USCOD aims to adaptively identify salient and camouflaged objects in unconstrained open scenarios, where each image may exist salient object, camouflaged object, both, or neither of them. Although our proposed CS12K dataset encompasses a wide array of scenarios and object categories, thereby enriching the learning experience for salient and camouflaged feature detection within unconstrained scenes for the COD and SOD communities, USCOD falls short in one critical aspect: it lacks the capability to differentiate between individual instances of salient and camouflaged objects. Specifically, USCOD is limited in recognizing the quantity of such instances and in distinguishing among different objects. This limitation undermines the effectiveness of algorithm in accurately discerning the unique camouflage and saliency patterns of each object. Moving forward, our research will venture into the domain of Unconstrained Salient and Camouflaged Instance Segmentation (USCIS), which strives to identify and segment each individual instance of salient and camouflaged objects. Figure 10 illustrates the label differences between USCOD and USCIS within the same image.

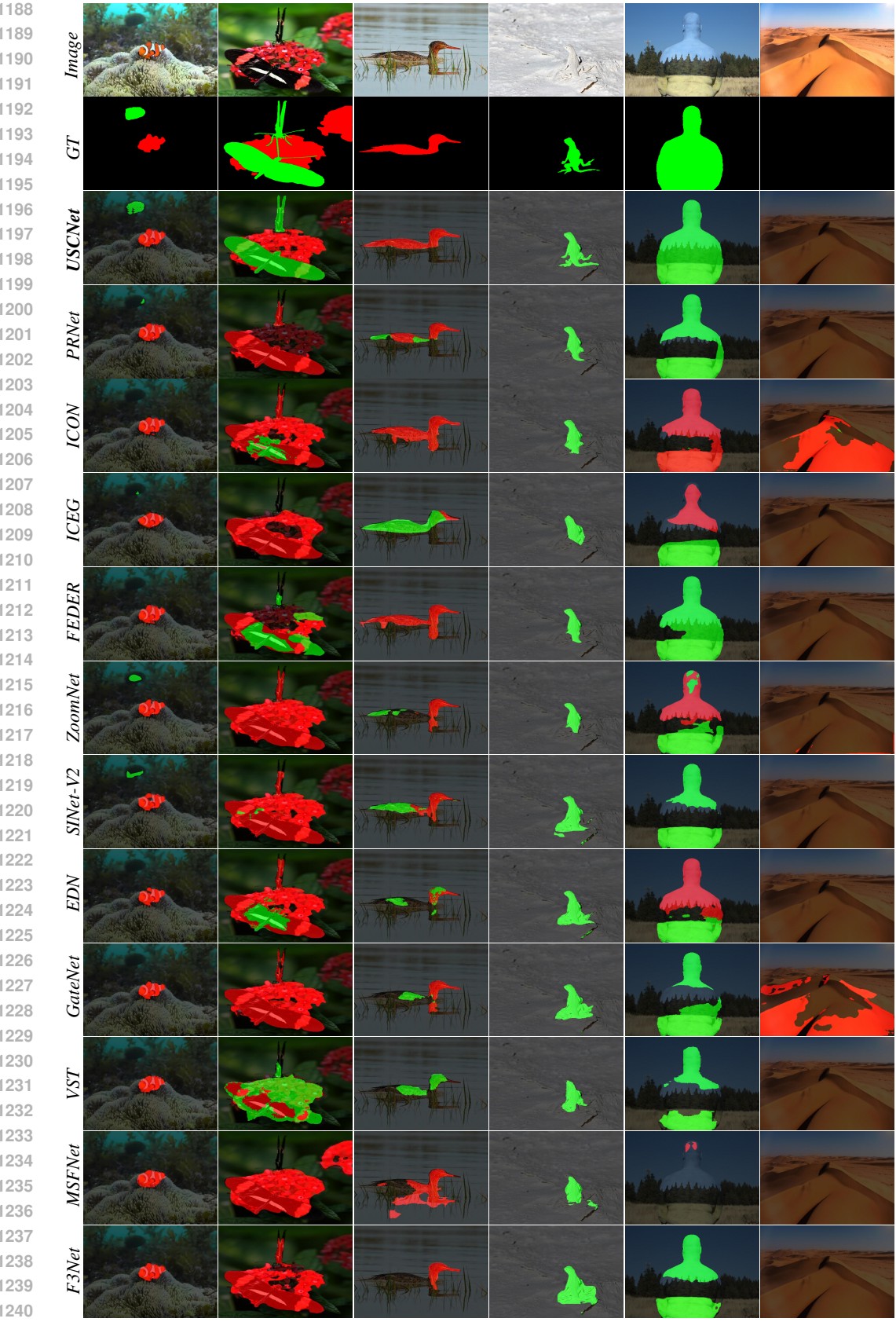

Figure 11: Additional visualizations of the proposed USCNet and other state-of-the-art methods on the CS12K test set. **Zoom-in for better view.**

