# OpenReview forum: "Unconstrained Salient and Camouflaged Object Detection"
_ICLR.cc/2025/Conference — Submitted to ICLR 2025_

### Official Review · Reviewer_LUqD · 2024-11-02

**Soundness:** 3
**Presentation:** 3
**Contribution:** 3
**Rating:** 8
**Confidence:** 5

**Summary:**

This paper introduces a new benchmark: USCOD, where both salient and camouflaged objects are considered during detection. Towards this, the authors construct a new dataset CS12K, containing four types of occasions, and proposes a new baseline for this benchmark. Furthermore, they design a new metric called CSCS to evaluate the performance of this task.

**Strengths:**

- The authors point out the current limitation of SOD and COD tasks, and introduce a new challenging task USCOD, combining SOD and COD together, which can improve the generalizability of both tasks.
- The authors carefully construct a new dataset for USCOD, which can contribute to this certain area.
- The competitive experiments demonstrate the essential to specially design datasets and networks for USCOD.

**Weaknesses:**

- More details about the CS12K are expected. For example, when manually annotating, how is the object judged to be as salient or camouflaged?
- The loss function of the proposed method is not stated clearly. In equation (4), is there some difference when calculating loss for salient and camouflaged objects? If not, the different difficulties of COD and SOD may make the optimization less effective.
- Also, when evaluating, there can be some factors influencing the actual performance. For example, the different difficulty of the two tasks: if the model performs well at SOD but poor at COD, will it achieve the same score on CSCS with another model which performs averagely at both tasks? Or will the size [1] and number [2] of objects influence the evaluation? The authors are expected to conduct some discussion on the evaluation.

[1] Li F, Xu Q, Bao S, et al. Size-invariance Matters: Rethinking Metrics and Losses for Imbalanced Multi-object Salient Object Detection. Proceedings of the 41st International Conference on Machine Learning, PMLR 235:28989-29021, 2024.

[2] Mingchen Zhuge, Deng-Ping Fan, Nian Liu, Dingwen Zhang, Dong Xu, and Ling Shao. Salient object detection via integrity learning. IEEE TPAMI, 45(3):3738–3752, 2022.

**Questions:**

1. How do you judge whether an object is salient or camouflaged? Does this process align with popular datasets?

2. When evaluating and optimizing, do you give equal treatment to salient and camouflaged objects?  If so, why not give different weights to the two types of objects considering their difference in difficulty?

3. In the metric CSCS, I did not see the $P_{CB}$ and $P_{SB}$, which reflects the false positive rate to some extent. Can you explain the reason why $P_{CB}$ and $P_{SB}$ are not considered?

---

> ### Author Response · Authors · 2024-11-25
> **Response to Reviewer LUqD (1/5)**
>
> We are honored and grateful to receive your positive evaluation. We sincerely appreciate your thorough review and the recognition of several strengths in our work, including the challenging new task, the valuable new dataset, and our essential network for USCOD. Hope we can address your concerns with the responses below.
>
> ---
>
> >***Q1:*** More details about the CS12K are expected. For example, when manually annotating, how is the object judged to be as salient or camouflaged?
> >
> >How do you judge whether an object is salient or camouflaged? Does this process align with popular datasets?
>
> ***Ans for Q1:***
>
> For Scene A and B, we randomly selected 3,000 images from existing public datasets. For Scene D, we collected 1,564 images from COD10K and manually collected 1,436 images from the internet. For Scene C, we manually filtered data with coexisting salient and camouflaged objects from existing public datasets, resulting in a total of 342 images, including 210 from COD10K, 40 from CAMO, 92 from NC4K, 32 from AWA2, and 9 from LUSI, while also collecting and filtering 2,617 images from the internet. The data sources are explained in detail in Section 3.1 of the manuscript. For Scene A and B, we retained their original annotations, while Scene D did not require additional annotation. Therefore, we focus here on **detailing the annotation process for Scene C**.
>
> - **Initial Determination of Object Attributes:** We invited 7 observers to perform the initial identification of salient and camouflaged objects in the images. A voting process was used to determine the salient and camouflaged objects in each image, with objects and their attributes receiving more than half of the votes being retained. We then used Photoshop to apply red boxes for salient objects and green boxes for camouflaged objects, which served as the reference for the subsequent mask annotation step.
>
> - **Mask Annotation:** We invited 9 volunteers to perform detailed mask annotation for the dataset using the ISAT interactive annotation tool *[1]*, which supports SAM semi-automatic labeling.
>
> - **Annotation Quality Control:** After annotation, we invited an additional 3 observers to review and refine the results. Masks with imprecise or incorrect annotations were manually corrected.
>
> In summary, we adopted a **voting and feedback mechanism** to ensure the quality of annotations, which aligns closely with the practices used in existing datasets *[2-5]*. Thank you for your suggestion. We will include more detailed information about dataset annotation in the Appendix of the revision.
>
> ---
> ---
>
> >***Reference:***
> >
> >*[1] https://github.com/yatengLG/ISAT_with_segment_anything*
> >
> >*[2] Lijun Wang, Huchuan Lu, Yifan Wang, Mengyang Feng, Dong Wang, Baocai Yin, and Xiang Ruan. Learning to detect salient objects with image-level supervision. In CVPR, 2017a.*
> >
> >*[3] Guanbin Li and Yizhou Yu. Visual saliency based on multiscale deep features. In CVPR, 2015.*
> >
> >*[4] Qiong Yan, Li Xu, Jianping Shi, and Jiaya Jia. Hierarchical saliency detection. In CVPR, 2013.*
> >
> >*[5] Deng-Ping Fan, Ge-Peng Ji, Guolei Sun, Ming-Ming Cheng, Jianbing Shen, and Ling Shao. Camouflaged object detection. In CVPR, pp. 2777–2787, 2020.*

---

> ### Author Response · Authors · 2024-11-25
> **Response to Reviewer LUqD (2/5)**
>
> >***Q2:*** The loss function of the proposed method is not stated clearly. In equation (4), is there some difference when calculating loss for salient and camouflaged objects? If not, the different difficulties of COD and SOD may make the optimization less effective.
> >
> >When evaluating and optimizing, do you give equal treatment to salient and camouflaged objects? If so, why not give different weights to the two types of objects considering their difference in difficulty?
>
> ***Ans for Q2:***
>
> Considering that **the number of images across the four scenes is balanced**, we assumed the proportions of salient and camouflaged objects to be consistent. Therefore, we treated salient and camouflaged objects equally during evaluation and optimization. Since the network already achieved state-of-the-art performance on existing metrics, no further optimization was conducted.
>
> However, your suggestion has been incredibly insightful. Inspired by your suggestion, we considered both the total pixel count and the difficulty of the two attributes, and attempted to address the issue by using **Focal Loss**. Specifically, we first calculated the pixel count ratio between salient and camouflaged objects, which is 6:4. Then, we set the weight ratio between saliency and camouflage attributes to 4:6, while keeping the default γ value of 2 to account for the varying difficulty of different pixels.
>
> The experimental results using different loss functions are summarized in *Table R1*:
>
> *Table R1: The Experimental Results using Different Loss Functions.*
> | Model | Loss | Scene A | Scene B | | Scene C | | | Overall | Scenes | | | |
> | -------- | -------- | -------- | -------- | -------- | -------- | -------- | -------- | -------- | -------- | -------- | -------- | -------- |
> | | | $IoU_S$↑ | $IoU_C$↑ | | $IoU_S$↑ $IoU_C$↑ | | | $IoU_S$↑ | $IoU_C$↑ | $mIoU$↑ | $mAcc$↑ | $CSCS$↓ |
> | **USCNet** | CE Loss | **79.70** | **74.99** | | 74.80 45.73 | | | **75.57** | **61.34** | **78.03** | 87.92 | 7.49 |
> | **USCNet** | Focal Loss | 78.73 | 73.28 | | **75.07 45.95** | | | 74.69 | 59.38 | 76.99 | **88.50** | **7.45** |
>
> From the table, we can observe that our model, when using Focal Loss, demonstrates potential, particularly in Scene C and in terms of CSCS. This suggests that **Focal Loss may help optimize the model’s ability to distinguish between salient and camouflaged objects in complex scenes**. However, no improvement in the IoU of Overall Scene was observed. We hypothesize that this lack of improvement could be due to suboptimal hyperparameter settings. We will continue to explore Focal Loss and other related methods to make the optimization more effective. Thank you again for your valuable suggestion!

---

> ### Author Response · Authors · 2024-11-25
> **Response to Reviewer LUqD (3/5)**
>
> >***Q3:*** Also, when evaluating, there can be some factors influencing the actual performance. For example, the different difficulty of the two tasks: if the model performs well at SOD but poor at COD, will it achieve the same score on CSCS with another model which performs averagely at both tasks?
>
> ***Ans for Q3:***
>
> To analyze the impact of the network's performance bias between the detection of salient and camouflaged objects on CSCS, we performed a detailed data analysis of the methods presented in Table 2 of the manuscript. The results are shown in *Table R2*. In this analysis, **Diff** represents the difference value between $IoU_S$ and $IoU_C$. **A smaller Diff indicates that the network treats the detection of the two attributes more equitably**, with less bias toward either attribute. We arranged the table from left to right based on ascending Diff values. It can be observed that **as Diff increases, CSCS does not exhibit a linear increase or decrease**. This indicates that CSCS is an independent metric focused on the degree of attribute confusion, and it does not vary systematically with the performance bias between the two attributes.
>
> *Table R2: The Analysis of the Impact of Diff on CSCS. Adapter denotes SAM2-Adapter.*
> | | USCNet | Adapter | ICEG | PRNet | EDN | EVP | ICON | SINetV2 | VSCode | MSFNet | GateNet | F3Net | PFNet | FEDER | ZoomNet | VST |
> |-------------|---------|--------------|--------|--------|--------|--------|--------|----------|--------|--------|---------|--------|--------|--------|---------|--------|
> |**$IoU_S$**| 75.57 | 71.42 | 62.72 | 68.68 | 68.00 | 70.30 | 65.86 | 67.50 | 67.09 | 66.69 | 65.08 | 66.12 | 65.73 | 68.65 | 66.43 | 63.18 |
> |**$IoU_C$**| 61.34 | 56.71 | 45.52 | 50.88 | 48.27 | 50.36 | 45.53 | 47.47 | 46.91 | 45.89 | 44.17 | 44.81 | 43.76 | 46.46 | 43.28 | 38.45 |
> |**Diff**|**14.23**|**14.71**|**17.20**|**17.80**|**19.73**|**19.94**|**20.33**|**20.03**|**20.18**|**20.80**|**20.91**|**21.31**|**21.97**|**22.19**|**23.15**|**24.73**|
> |**CSCS**| 7.49 | 9.12 | 8.61 | 8.40 | 9.23 | 8.67 | 10.24 | 8.83 | 9.72 | 9.90 | 11.30 | 9.36 | 10.04 | 10.01 | 8.88 | 11.30 |

---

> ### Author Response · Authors · 2024-11-25
> **Response to Reviewer LUqD (4/5)**
>
> >***Q4:*** Also, when evaluating, there can be some factors influencing the actual performance. Will the size [1] and number [2] of objects influence the evaluation? The authors are expected to conduct some discussion on the evaluation.
>
> ***Ans for Q4:***
>
> Additionally, our USCOD benchmark adopts pixel-level evaluation metrics rather than instance-level metrics (like the Average Precision (AP) metric in the COCO benchmark) or sample-level metrics (like the F-measure widely used in the SOD and COD fields, which calculates the F-measure for each sample). Instead, we adopt an approach inspired by semantic segmentation benchmarks (such as Cityscapes, PASCAL VOC, and ADE20K) to comprehensively evaluate the model’s performance in recognizing saliency, camouflage, and background attributes.
>
> Specifically, we calculate **a confusion matrix across all pixels** in the testing set, followed by calculating the IoU score for each attribute, as well as the mIoU, mAcc, and our CSCS metric. **Therefore, the evaluation is not influenced by the size *[1]* or number *[2]* of objects**. The Appendix of the manuscript provides the confusion matrix generated by our method on the CS12K testing set.
>
> However, after carefully reviewing *[1]* and *[2]*, we realized that the size and number of objects are crucial factors in evaluating Multi-object Salient Object Detection. Similarly, for our proposed USCOD task, this is equally important since multi-object scenarios are widely represented in our benchmark.
>
> To analyze the impact of object size and number on the model's performance in USCOD, we **adopted the metrics mentioned in *[1]*** to evaluate our model's performance in detecting salient and camouflaged object. Additionally, we assessed the performance of the SOD model ICON and the COD model ICEG for comparison. The detailed results are shown in *Table R3* and *Table R4*. It can be observed that, compared to the size-sensitive (e.g., $AUC↑$ and $F_m^{\beta}↑$) and size-invariance (e.g., SI-$AUC↑$ and SI-$F_m^{\beta}↑$) metrics, **our method exhibits smaller performance fluctuations, demonstrating its robustness to variations in object size and number in the scene**.
>
> *Table R3: Performance of detecting salient object on CS12K testing set.*
> | Model | $AUC↑$ | SI-$AUC↑$ | $F_m^{\beta}↑$ | SI-$F_m^{\beta}↑$ | $F_{\max}^{\beta}$↑ | SI-$F_{\max}^{\beta}$↑ | Em↑ |
> |--------|--------|----------|--------|----------|---------|----------|---------|
> | ICON | 0.8207 | 0.8320 | 0.7017 | 0.7644 | 0.7105 | 0.7739 | 0.7951 |
> | ICEG | 0.8301 | 0.8253 | 0.7343 | 0.7616 | 0.7429 | 0.7704 | 0.8308 |
> | **Ours** | **0.8531** | **0.8500** | **0.7605** | **0.7868** | **0.7721** | **0.7980** | **0.8327** |
>
> *Table R4: Performance of detecting camouflaged object on CS12K testing set.*
> | Model | $AUC↑$ | SI-$AUC↑$ | $F_m^{\beta}↑$ | SI-$F_m^{\beta}↑$ | $F_{\max}^{\beta}$↑ | SI-$F_{\max}^{\beta}$↑ | Em↑ |
> |--------|--------|----------|--------|----------|--------|-----------|---------|
> | ICON | 0.6625 | 0.6632 | 0.3836 | 0.5489 | 0.3941 | 0.5605 | 0.5866 |
> | ICEG | 0.7306 | 0.7170 | 0.5249 | 0.6009 | 0.5320 | 0.6089 | 0.7194 |
> | **Ours** | **0.8014** | **0.7935** | **0.6100** | **0.6584** | **0.6185** | **0.6671** | **0.7949** |
>
> To comprehensively evaluate these important metrics, we will **incorporate them into the USCOD benchmark and continue assessing other models' performance on these metrics**. Thank you for your valuable suggestion.
>
> In the future, we also plan to explore an instance-level USCOD detection task, namely the **Unconstrained Salient and Camouflaged Instance Segmentation (USCIS)** task. The details of this task can be found in §G of the Appendix in the manuscript. For USCIS, the size-invariant metrics and size-invariant optimization loss mentioned in *[1]* are likely to be more suitable for evaluating and optimizing model performance. Once again, we sincerely appreciate your insights! If you have any further suggestions, we would be most grateful to hear them.
>
> ---
> ---
>
> >***Reference:***
> >
> >*[1] Li F, Xu Q, Bao S, et al. Size-invariance Matters: Rethinking Metrics and Losses for Imbalanced Multi-object Salient Object Detection. Proceedings of the 41st International Conference on Machine Learning, PMLR 235:28989-29021, 2024.*
> >
> >*[2] Mingchen Zhuge, Deng-Ping Fan, Nian Liu, Dingwen Zhang, Dong Xu, and Ling Shao. Salient object detection via integrity learning. IEEE TPAMI, 45(3):3738–3752, 2022.*

---

> ### Author Response · Authors · 2024-11-25
> **Response to Reviewer LUqD (5/5)**
>
> >***Q5:*** In the metric CSCS, I did not see the $P_{CB}$ and $P_{SB}$, which reflects the false positive rate to some extent. Can you explain the reason why $P_{CB}$ and $P_{SB}$ are not considered?
>
> ***Ans for Q5:***
>
> The CSCS calculation formula consists of two components: the proportion of camouflage pixels incorrectly predicted as saliency among all pixels predicted as saliency, and the proportion of saliency pixels incorrectly predicted as camouflage among all pixels predicted as camouflage, which is inspired by IoU. The evaluation focuses on the model's ability to distinguish between saliency and camouflage pixels, specifically assessing the degree of confusion between these two attributes, without considering the salient and camouflaged pixels incorrectly predicted as background, i.e., $P_{SB}$ and $P_{CB}$.

---

> > ### Comment · Reviewer_LUqD · 2024-11-26
> >
> > Thank you for the responses. The authors have thoroughly addressed my concerns, and I appreciate the comprehensive explanations and additional experiments provided for all my questions. Additionally, the APG module appears well-suited for this task, as evidenced by the responses to other reviewers' questions.
> >
> > Before making the final rating, I still expect the authors to name some practical applications when introducing USCOD, as these could better illustrate the necessity and significance of this new task.

---

> > > ### Author Response · Authors · 2024-11-29
> > > **Response to Reviewer LUqD**
> > >
> > > Thank you for your valuable suggestions! We have added some typical practical applications of USCOD in §G of the Appendix in the revision to better illustrate the necessity and significance of this new task.
> > >
> > > ---
> > >
> > > - **Military Surveillance and Enemy Reconnaissance:** In a military environment, salient objects might include large military equipment such as vehicles, tanks, helicopters, etc., while camouflaged objects could be soldiers or equipment hidden in vegetation or camouflage materials. The simultaneous detection of both salient and camouflaged objects helps enhance battlefield situational awareness and prevents overlooking potential threats.
> > >
> > > - **Post-Disaster Search and Rescue:** After a disaster, salient objects might include obvious signs of life in rubble (such as clearly visible trapped individuals), while camouflaged objects could be life signs that are difficult to detect due to obstruction or chaotic environments (such as partially buried survivors). The simultaneous detection of both salient and camouflaged objects is crucial for improving search and rescue efficiency.
> > >
> > > - **Ecological Protection and Wildlife Monitoring:** In natural environments, salient objects might be easily visible animals (such as birds in open areas), while camouflaged objects could be animals hidden in vegetation (such as insects with protective coloration). The simultaneous detection of both salient and camouflaged objects allows for more comprehensive wildlife population surveys and ecological research.
> > >
> > > - **Multi-Level Lesion Detection:** In medical imaging, detecting both salient lesions (such as obvious tumors or organ damage) and camouflaged lesions (such as those blurred by background textures or early-stage lesions) helps doctors more thoroughly assess a patient's health condition.
> > >
> > > - **Diving Hazard Warnings:** During diving, salient objects might include coral or schools of fish that attract the diver’s attention, while camouflaged objects could be hidden dangerous creatures (such as stonefish or moray eels). The simultaneous detection of both salient and camouflaged objects helps guide divers in more comprehensively avoiding dangers.
> > >
> > > Additionally, we have incorporated the evaluation metrics from *[1]* into the USCOD benchmark and evaluated the performance of other models on these metrics. The complete evaluation results can be found in Table 4 and Table 5 in Appendix §B of the revision we have submitted.
> > >
> > > We greatly appreciate your valuable feedback and the insightful papers you provided, which are closely related to our work and have been extremely helpful to us.
> > >
> > > ---
> > > ---
> > >
> > > >***Reference:***
> > > >
> > > >*[1] Li F, Xu Q, Bao S, et al. Size-invariance Matters: Rethinking Metrics and Losses for Imbalanced Multi-object Salient Object Detection. Proceedings of the 41st International Conference on Machine Learning, PMLR 235:28989-29021, 2024.*

---

> > > > ### Comment · Reviewer_LUqD · 2024-11-29
> > > >
> > > > The authors have successfully demonstrated the necessity of this new task. I appreciate their efforts in introducing new datasets, evaluation metrics, and an effective solution. Consequently, I have decided to raise my rating.

---

### Official Review · Reviewer_FwYu · 2024-11-03

**Soundness:** 2
**Presentation:** 2
**Contribution:** 2
**Rating:** 5
**Confidence:** 3

**Summary:**

This paper addresses both salient object detection (SOD) and camouflaged object detection (COD). Specifically, the authors first introduce a benchmark called Unconstrained Salient and Camouflaged Object Detection (USCOD) and construct a large database, CS12K. They then propose a SAM-based baseline to tackle USCOD.

**Strengths:**

1. The construction of the CS12K dataset has contributed to the further development of this task.
2. This paper proposes a new metric for simultaneously evaluating saliency and camouflage detection.
3. The writing is good and easy to read.

**Weaknesses:**

1. Is the comparison with other models fair? Were the other models compared using the same training data as this model?
2. The model design in this paper does not look very novel. Specifically, this paper proposes an adapter prompting strategy based on a cross-attention mechanism, which is very similar to [1].
3. The authors repeatedly state that distinguishing salient and camouflaged objects within the same image is challenging. However, theoretically, as salient and camouflaged are antonyms, it should not be excessively difficult to differentiate between them.

[1] Sun, Yanpeng, et al. "VRP-SAM: SAM with visual reference prompt." Proceedings of the IEEE/CVF Conference on Computer Vision and Pattern Recognition. 2024.

**Questions:**

1. Is the comparison with other models fair? Were the other models compared using the same training data as this model?

---

> ### Author Response · Authors · 2024-11-21
> **Response to Reviewer FwYu (1/2)**
>
> Thank you for your valuable time and thoughtful feedback. We greatly appreciate your recognition of our dataset, new metric, and the readability of our manuscript. Below are our responses to the concerns you raised, and we hope that these answers effectively address your questions.
>
> ---
>
> >***Q1:*** Is the comparison with other models fair? Were the other models compared using the same training data as this model?
>
> ***Ans for Q1:***
>
> We describe the training set and testing set used for the comparison experiments **in L402-403 of the manuscript**. Before evaluating the performance of each model, we retrain **all the models using the same training set** (CS12K training set with 8,400 images) as our USCNet to ensure fairness in the evaluation.
>
> - After training, we evaluate all 16 models using the CS12K testing set, which consists of 3,600 images covering all scenes. We use mIoU, mAcc, and CSCS as evaluation metrics. The experimental results are shown in Table 2.
>
> - Additionally, to further validate the effectiveness and robustness of our model regarding generalizability, we conduct tests on popular SOD datasets (DUTS, HKU-IS, and DUT-OMRON) and COD datasets (CAMO, COD10K, and NC4K), with all models uniformly trained using our CS12K training set. In these experiments, we adopt five metrics that are widely used in COD and SOD tasks: maximum F-measure ($F_β^{max}$ ↑), weighted F-measure ($F_β^{ω}$ ↑), Mean Absolute Error (MAE, M ↓), Structural measure (S-measure, $S_α$ ↑), and mean Enhanced alignment measure (E-measure, $E_ϕ^m$ ↑). This part of the experiment is described in *Appendix §B* of the manuscript.
>
> ---
>
> >***Q2:*** The model design in this paper does not look very novel. Specifically, this paper proposes an adapter prompting strategy based on a cross-attention mechanism, which is very similar to [1].
>
> ***Ans for Q2:***
>
> We have carefully reviewed the article *[1]* you provided and conducted a study on VRP-SAM. While both our model and VRP-SAM generate adapter prompts for SAM, the **processes and objectives behind prompt generation** in our approach differ significantly from those in VRP-SAM.
>
> **1. Differences in the Prompt Generation Process**
>
> - In VRP-SAM, a set of learnable queries $Q ∈ R^{N×C}$ is introduced, and two rounds of cross-attention and self-attention are performed. In the first round, $Q$ interacts with the visual features of the reference image to obtain $Q'_r ∈ R^{N×C}$. In the second round, $Q'_r$ interacts with the target image to generate $Q'_t ∈ R^{N×C}$, which ultimately serve as the visual reference prompt embeddings for SAM.
>
> - Compared to VRP-SAM, to obtain discriminative **S**aliency, **C**amouflage and **B**ackground prompts for the USCOD task, our model employs three sets of static learnable queries {$Q_{S-S} ∈ R^{N×C}, Q_{S-C} ∈ R^{N×C}, Q_{S-B} ∈ R^{N×C}$} and three sets of dynamic queries {$Q_{D-S} ∈ R^{N×C}, Q_{D-C} ∈ R^{N×C}, Q_{D-B} ∈ R^{N×C}$} generated from image feature. Static queries focus on capturing general attribute features that remain consistent across samples, regardless of the input during inference. Unlike the static queries, dynamic queries are derived from the input image, enabling them to capture unique attribute features specific to the given sample. By integrating static and dynamic prompt queries, we construct three attribute prompt queries: {$Q_S ∈ R^{N×C}, Q_C ∈ R^{N×C}, Q_B ∈ R^{N×C}$}. These queries are concatenated into a unified prompt query $Q ∈ R^{3×N×C}$. Then, the combined query leverages self-attention to capture differences among the three attribute prompts, followed by cross-attention to integrate input image features and enhance their learning.
>
> **2. Differences in Optimization Objectives**
>
> - The Prompt Generator in VRP-SAM is designed to learn the visual features of the reference image, generating **a general, class-specific visual semantic prompt**.
>
> - In comparison, our APG module extracts attribute features from the input image, generating **three distinct, class-agnostic attribute prompts**.
>
> Thank you for providing a relevant work. We will include a discussion of VRP-SAM in the related work of the revision.
>
> >**Reference:**
> >
> >*[1] Sun, Yanpeng, et al. "VRP-SAM: SAM with visual reference prompt." Proceedings of the IEEE/CVF Conference on Computer Vision and Pattern Recognition. 2024.*

---

> ### Author Response · Authors · 2024-11-21
> **Response to Reviewer FwYu (2/2)**
>
> >***Q3:*** The authors repeatedly state that distinguishing salient and camouflaged objects within the same image is challenging. However, theoretically, as salient and camouflaged are antonyms, it should not be excessively difficult to differentiate between them.
>
> ***Ans for Q3:***
>
> Although saliency and camouflage are conceptually opposites, distinguishing between salient and camouflaged objects is not straightforward in complex real-world scenarios. When both of them coexist within the same scene, they may have **similar visual features** or **highly overlapping regions**, increasing the difficulty of differentiating between them.
>
> - When salient and camouflaged objects have **similar visual features**, the model cannot rely on a single characteristic to make the distinction. Instead, it must consider multiple visual cues within the scene. For example, in Fig. 8 (Row 1, Column 3) of the manuscript, two leopards appear together. Although their shapes are similar, they belong to different attributes—one is salient, and the other is camouflaged. In this case, the model must analyze local texture variations, surrounding differences, lighting conditions, and other multi-dimensional features to accurately differentiate between the two objects.
>
> - When salient and camouflaged objects **overlap or are in close proximity**, the salient object may distract the model and interfere with its ability to perceive the camouflaged object. For example, in Fig. 8 (Row 5, Column 4) of the manuscript, a yellow spider is hidden within a yellow flower. The flower is the salient object, while the spider is camouflaged. The spider's camouflaged nature arises because it overlaps with the flower. In this case, the model needs to deeply analyze the contextual relationship between the two objects in order to accurately distinguish them.
>
> **This challenge is also reflected in our experimental results.** As shown in Table 2 of the manuscript, the $IoU_S$ for salient objects in Scene C is, on average, 6.25% lower compared to Scene A, and the $IoU_C$ for camouflaged objects in Scene C is, on average, 25% lower compared to Scene B. These results highlight the challenge of distinguishing between salient and camouflaged objects in scenes that contain both.
>
> Therefore, although these two terms are opposites, their distinction becomes blurred in complex, unstructured scenes, significantly increasing the difficulty of differentiating between them. To effectively address this challenge, the model must possess sufficient flexibility to learn and differentiate these objects in unconstrained scenes.

---

> > ### Comment · Reviewer_FwYu · 2024-11-23
> >
> > Thank you for your responses. While the authors addressed my concerns regarding comparison fairness and task difficulty, I still maintain my previous judgment regarding novelty.  Although they claim to introduce static and dynamic learnable queries, I think the only difference between these two sets of queries lies in whether the tokens are initialized randomly or based on samples, so the method in this paper is not substantially different from VRP-SAM. I will keep my original rating.

---

> > > ### Author Response · Authors · 2024-11-29
> > > **Response to Reviewer FwYu**
> > >
> > > Thank you for your careful reading. In fact, many works have adopted the adapter prompting strategy, which is currently widely applied across various tasks. The novelty of these works lies more in **how they generate prompts that are more suitable for specific tasks**, rather than the strategy itself.
> > >
> > > **1. VRP-SAM *[1]***: Uses queries to generate visual reference prompts, but what is more relevant to the **Visual Reference Segmentation task** is the use of a **Feature Augmenter** to extract and enhance the semantic correlation between the reference image and the target image, providing accurate feature representations.
> > >
> > > **2. RSPrompter *[2]***: The RSPrompter-query uses queries to obtain prompt embeddings related to the semantic category, but what is more relevant to the **Remote Sensing Instance Segmentation task** is the use of **Optimal Transport Matching** to optimize the alignment between the predicted instance masks and the ground truth masks.
> > >
> > > **3. SurgicalSAM *[3]***: Uses queries to obtain category-related prompts, but what is more relevant to the **Surgical Instrument Segmentation task** is the introduction of a **Prototype-based Class Prompt Encoder** and Contrastive Prototype Learning, which enhance the distinguishability between category prototypes and improve the ability to recognize fine-grained surgical instrument categories.
> > >
> > > **4. QaP *[4]***: Uses queries to obtain modality-specific prompts that interact with text information, but what is more relevant to the **Multimodal Language Model task** is the **Text-Conditioned Resampler**, which can adaptively extract text-conditioned information from different modalities (audio or vision).
> > >
> > > In comparison, our **USCNet** uses queries to generate attribute-specific prompts, but for the **USCOD task**, the more critical aspect is **how to generate discriminative prompts** to distinguish salient and camouflaged objects within the same scene.
> > >
> > > Our insight is that distinguishing salient and camouflaged objects within the same scene requires considering two dimensions: **Sample-generic features**, such as size, position, color, and texture, which are important universal features for distinguishing salient and camouflaged objects, and **Sample-specific features**, which are closely related to the current sample when salient and camouflaged objects share similar features (e.g., colors or categories). To capture features from different dimensions, we integrate two types of queries in the APG module: **Static Prompt Query (SPQ)** and **Dynamic Prompt Query (DPQ)**, which are designed to capture Sample-generic features and Sample-specific features, respectively. As Reviewer LUqD mentioned, our APG module is suitable for the USCOD task. Integrating the two types of prompts results in more discriminative salient and camouflage prompts, which better distinguish salient and camouflaged objects within the same scene, thus addressing the challenges of the USCOD task more effectively.
> > >
> > > Overall, the novelty of our network lies in the design of the APG module based on this insight, rather than the use of the Query as Prompt paradigm, which is widely used in other tasks, including VRP-SAM, RSPrompter, SurgicalSAM, and QaP.
> > >
> > > We will clarify the distinction between our work and the aforementioned related works in the revision. If you have any further concerns, please feel free to let us know, and we will do our best to address any issues you may have.
> > >
> > > ---
> > > ---
> > >
> > > >***Reference:***
> > > >
> > > >*[1] Sun, Yanpeng, et al. "VRP-SAM: SAM with visual reference prompt." Proceedings of the IEEE/CVF Conference on Computer Vision and Pattern Recognition. 2024.*
> > > >
> > > >*[2] Chen K, Liu C, Chen H, et al. RSPrompter: Learning to prompt for remote sensing instance segmentation based on visual foundation model[J]. IEEE Transactions on Geoscience and Remote Sensing, 2024.*
> > > >
> > > >*[3] Yue W, Zhang J, Hu K, et al. Surgicalsam: Efficient class promptable surgical instrument segmentation[C]//Proceedings of the AAAI Conference on Artificial Intelligence. 2024, 38(7): 6890-6898.*
> > > >
> > > >*[4] Liang T, Huang J, Kong M, et al. Querying as Prompt: Parameter-Efficient Learning for Multimodal Language Model[C]//Proceedings of the IEEE/CVF Conference on Computer Vision and Pattern Recognition. 2024: 26855-26865.*

---

> ### Author Response · Authors · 2024-12-03
> **Reminder for Approaching Deadline**
>
> Dear Reviewer FwYu,
>
> The deadline for posting messages to authors is approaching, and we may soon be unable to view your new comments. If you have any further concerns, please feel free to let us know, and we will do our best to address them before the deadline for posting messages to reviewers.
>
> Thank you again for your valuable time and feedback.
>
> Best regards,
>
> Authors of #38

---

### Official Review · Reviewer_224k · 2024-11-04

**Soundness:** 3
**Presentation:** 3
**Contribution:** 1
**Rating:** 3
**Confidence:** 5

**Summary:**

This work aims to solve the unconstrained salient and camouflaged object detection problem. It is a challenging task, that needs to jointly detect the potential salient and/or camouflaged objects in the image. To this end, they propose the first joint salient and camouflaged object detection dataset, CS12K. They also propose a solution, USCNet, for the task. The experimental results shown their effectiveness, in particularly in scenes containing both salient and camouflaged objects.

**Strengths:**

1. They propose a great USCOD dataset, CS12K. Different from previous SOD/COD dataset, CS12K includes images containing both salient and camouflaged objects jointly. It is more challenging and may be practical in some scenes.

2. They conduct sufficient experiments to proof the effectiveness of their approach.

**Weaknesses:**

1. The motivation of this work is unclear. I am curious that why we need to jointly detect salient and camouflaged objects without any pre-defined prompts? I cannot see critical problems if using the pre-defined prompts. May the authors add more explanations regarding this?

2. The novelty of this work is very limited. I cannot see the technique novelty. One APG is proposed, but the APG design is not novel, since using different type of prompts for different task setting is usual. The idea of the proposed method is not novel as well. May the author highlight it?

3. The contribution of this work is insufficient. While it proposes a great dataset, the motivation is unclear, and the idea and method are not novel to me. I believe the contribution is not enough.

**Questions:**

1. In L088-L091, "The visual prompt of EVP, as part of the model, needs to be retrained according to the datasets of different tasks and requires pre-defining the category of the detection task. Similarly, the task prompts in VSCode and Spider also need to be pre-defined. Similarly, the task prompts in VSCode and Spider also need to be pre-defined." The "Similarly" here is confusing. Do VSCode and Spider need to be retrained? Or only require pre-defined prompts? Thanks for classify this!

2. If VSCode and Spider do not need to be retrained, I am curious about their evaluation protocols, as well as that of EVP (Liu 2023). How are the prompts set for scenes A, B, C, and D? I cannot find any details regarding this, but I believe it is critical since it directly relates to the fairness and necessity of using the same type of prompt for the USCOD task. Thus, I hope the authors can make the evaluation protocols clearer (this is important).

3. Regarding the CSCS metrics, based on the formulation, it seems they only consider the confusion between salient and camouflaged pixels, while the false positives and false negatives between salient/camouflaged pixels and the background are not considered. Therefore, we still need to look at mIOU to understand the quality of the method. However, mIOU can also reveal aspects such as the confusion between salient and camouflaged pixels. Thus, I wonder why we still need CSCS metrics?

4. If the paper is accepted, will the dataset be released in its “full version”? By this, I mean the version containing the entire training set (8,400 images and the corresponding masks) and the testing set (3,600 images and the corresponding masks), along with the scene types.

---

> ### Author Response · Authors · 2024-11-20
> **Response to Reviewer 224k (1/2)**
>
> We sincerely thank you for your time and insightful comments. We also appreciate your recognition of our dataset and the sufficiency of our experiments. Below are our responses to your concerns, and we hope that these answers address them effectively.
>
> ---
>
> >***Q1:*** The motivation of this work is unclear. I am curious that why we need to jointly detect salient and camouflaged objects without any pre-defined prompts? I cannot see critical problems if using the pre-defined prompts. May the authors add more explanations regarding this?
>
> ***Ans for Q1:***
>
> The motivation of this work is to solve the problem of **jointly detecting salient and camouflaged objects in more complex and unconstrained scenes**.
>
> - Previous methods that **required pre-defined prompts** unify and integrate two independent binary classification tasks (SOD and COD). These methods work effectively in simplified scenes where only salient or camouflaged objects are present. However, in more complex scenarios where **salient and camouflaged objects coexist** within the same scene, they face limitations. Specifically, to jointly detect objects in such scenarios, these methods require two separate prompts for detecting each attribute, performing two binary classifications, and generating two binary masks. When merging the masks to obtain the final prediction, the **overlap of pixels** between the two masks can **introduce uncertainty into the prediction result**.
>
> - In contrast, by jointly detecting both types of objects **without pre-defined prompts**, our model can learn to differentiate these objects more flexibly and robustly. To be specifically, the final predictions are generated internally by the model, eliminating the need for post-processing. This approach enables the model to **learn the attributes of uncertain pixels during training**, effectively addressing the issue of pixel overlap. In this way, our model is able to make more accurate predictions **in more realistic and complex scenarios**.
>
> ---
>
> >***Q2:*** The novelty of this work is very limited. I cannot see the technique novelty. One APG is proposed, but the APG design is not novel, since using different type of prompts for different task setting is usual. The idea of the proposed method is not novel as well. May the author highlight it?
>
> ***Ans for Q2:***
>
> We understand your concern regarding the perceived novelty of our method. Using different types of prompts for different tasks is indeed a common approach. However, in these methods, **the prompts are independent** and merely serve to distinguish between tasks, **with no interaction or learning between them**. This can result in the model being unable to learn sufficiently discriminative prompts for the USCOD task.
>
> - In contrast, our APG module focuses on generating distinct and discriminative prompts for the three attributes (saliency, camouflage, and background) within the same scene. It **models the interrelationships between the prompts of three attributes**, thereby generating **discriminative attribute prompts** that better differentiate the salient and camouflaged patterns of objects.
>
> - Additionally, our architecture **decouples attribute distinction from mask reconstruction** by freezing the SAM mask decoder, which allows the network to **focus on learning the attributes** in a more targeted manner.
>
> This combination of **modeling attribute relationships** and **decoupling classification from mask reconstruction** is what sets our approach apart from existing techniques and contributes to its novelty.
>
> ---
>
> >***Q3:*** In L088-L091, "The visual prompt of EVP, as part of the model, needs to be retrained according to the datasets of different tasks and requires pre-defining the category of the detection task. Similarly, the task prompts in VSCode and Spider also need to be pre-defined. Similarly, the task prompts in VSCode and Spider also need to be pre-defined." The "Similarly" here is confusing. Do VSCode and Spider need to be retrained? Or only require pre-defined prompts? Thanks for classify this!
>
> ***Ans for Q3:***
>
> To clarify, the *"retrain"* in **EVP** refers to the need to train different visual prompts multiple times based on the datasets of different tasks, resulting in task-specific visual prompts. In contrast, for **VSCode** and **Spider**, during training, it is only necessary to distinguish between the prompts across datasets of different tasks, and combine multiple datasets of different tasks for training once. The term *"Similarly"* in this context refers to the fact that *"task prompts need to be pre-defined"* during inference, rather than retraining the model. Sorry for the confusion and we will revise the wording in the revision accordingly.

---

> > ### Comment · Reviewer_224k · 2024-11-22
> > **Thank you for the response!**
> >
> > Thank you for the response! I have read other reviews and authors' response. I think the novelty and contribution of this work cannot meet the requirement of ICLR. By the way, I think Reviewer Htmz raised some important questions that have not been addressed. Thus, I am lowering my score.

---

> ### Author Response · Authors · 2024-11-20
> **Response to Reviewer 224k (2/2)**
>
> >***Q4:*** If VSCode and Spider do not need to be retrained, I am curious about their evaluation protocols, as well as that of EVP (Liu 2023). How are the prompts set for scenes A, B, C, and D? I cannot find any details regarding this, but I believe it is critical since it directly relates to the fairness and necessity of using the same type of prompt for the USCOD task. Thus, I hope the authors can make the evaluation protocols clearer (this is important).
>
> ***Ans for Q4:***
>
> In order to ensure fairness in the evaluation, we **retrain all models** to be evaluated on the **training set of CS12K (8,400 images)**, as detailed in L402-L403 of the manuscript.
>
> - For the unified models, **VSCode**, **Spider**, and **EVP**, which require task-specific prompts for each dataset, we create two copies of the CS12K training set. One copy is used for SOD, with the ground truth being the SOD-only mask, and is used to train the prompts corresponding to the SOD task. The other copy is used for COD, with the ground truth being the COD-only mask, and is used to train the prompts corresponding to the COD task.
>
> - VSCode and Spider are **trained once** using all 16,800 images (two copies of 8,400 images), while EVP is **trained twice** on the two separate training sets (each containing 8,400 images) to obtain the two task-specific prompts.
>
> - During **evaluation**, all unified models perform inference on the testing set of CS12K twice, with the corresponding prompt enabled for each task. The first inference run generates the SOD results, and the second inference run generates the COD results. The final prediction is obtained by **merging the SOD and COD predictions**. For overlapping pixels, the attribute with the higher prediction value between the two tasks is chosen as the final attribute for that pixel.
>
> Thank you very much for your careful review. We apologize for not providing a clear description of the evaluation details, and we will provide a more detailed explanation in the revision.
>
> ---
>
> >***Q5:*** Regarding the CSCS metrics, based on the formulation, it seems they only consider the confusion between salient and camouflaged pixels, while the false positives and false negatives between salient/camouflaged pixels and the background are not considered. Therefore, we still need to look at mIOU to understand the quality of the method. However, mIOU can also reveal aspects such as the confusion between salient and camouflaged pixels. Thus, I wonder why we still need CSCS metrics?
>
> ***Ans for Q5:***
>
> The CSCS metric is specifically designed to assess the degree of confusion between saliency and camouflage, which is a key aspect of model evaluation. While mIoU is indeed a valuable metric for evaluating a method's ability to distinguish saliency, camouflage, and background, CSCS provides **a more focused evaluation** of how well the model **distinguishes between salient and camouflaged objects**.
>
> - In other words, CSCS allows us to directly measure the accuracy of distinguishing between these two attributes of objects. mIoU, while useful, does not fully capture this aspect, as it also accounts for the background class and may dilute the specific performance in distinguishing between salient and camouflaged objects.
>
> - In many cases, it is necessary to combine both metrics for a more detailed evaluation of model performance. For example, in Table 2 of the manuscript, PFNet and ZoomNet have similar mIoU scores, but there is a noticeable difference in their CSCS values.
>
> Therefore, **both CSCS and mIoU are complementary metrics**, with CSCS offering a more direct measure of performance on the salient/camouflaged distinction, and mIoU providing a broader view of overall segmentation quality.
>
> ---
>
> >***Q6:*** If the paper is accepted, will the dataset be released in its “full version”? By this, I mean the version containing the entire training set (8,400 images and the corresponding masks) and the testing set (3,600 images and the corresponding masks), along with the scene types.
>
> ***Ans for Q6:***
>
> Sure, if the paper is accepted, we guarantee that the complete dataset will be released for related domain research, including the entire training set (8,400 images and corresponding masks), the test set (3,600 images and corresponding masks), along with the scene types. We are committed to advancing research in the related field, and it would be an honor to contribute to its development.

---

> ### Author Response · Authors · 2024-11-25
> **Thank you for your feedback and looking forward to your further review.**
>
> Dear Reviewer 224k,
>
> We apologize for not being able to respond to all reviewers' comments simultaneously. We wanted to respond to each reviewer’s comments one by one to ensure the comprehensiveness of our responses.
>
> In the past few days, we have made every effort to carefully address each reviewer’s concerns, providing explanations and conducting additional experiments. Since Reviewer Htmz's comments included some additional experiments, we only responded after confirming that we had completed all the necessary experiments.
>
> We have now provided our first round of responses to all reviewers’ comments, including all of Reviewer Htmz's comments. We kindly ask you to review our responses again, and we are committed to addressing any other issues or concerns you may have.
>
> Thank you once again for your valuable feedback.
>
> Best regards,
>
> Authors of #38

---

> > ### Comment · Reviewer_224k · 2024-11-29
> > **Thanks for the response**
> >
> > Thanks for the response! I have read the comments and responses. I suggest the authors can include the additional experiments (added during the discussion period) and highlight their novelty and contribution in their next submission. This should significantly improve their work. However, I think the novelty of current submission is limited, and contribution is not sufficient, thus, I maintain my rating. Thanks!

---

> > > ### Author Response · Authors · 2024-12-03
> > > **Reminder for Approaching Deadline**
> > >
> > > Dear Reviewer 224k,
> > >
> > > We greatly appreciate all the reviewers' suggestions during the rebuttal period. In fact, we have already made modifications in the submitted revision, including additional experiments and more detailed explanations.
> > >
> > > The deadline for posting messages to authors is approaching, and we may soon be unable to view your new comments. If you have any further concerns or more detailed suggestions, please feel free to let us know, and we will do our best to address them before the deadline for posting messages to reviewers.
> > >
> > > Thank you again for your valuable time and feedback.
> > >
> > > Best regards,
> > >
> > > Authors of #38

---

### Official Review · Reviewer_Htmz · 2024-11-05

**Soundness:** 2
**Presentation:** 2
**Contribution:** 2
**Rating:** 3
**Confidence:** 4

**Summary:**

This work proposes a USCOD benchmark to bridge the difference between salient and camouflaged object detection. The authors explains that, although there are many works highlighting COD and SOD individually, USCOD tries to generalize the scenario by detecting both camouflaged and salient objects. To achieve that, the authors propose a new dataset CS12K, a new evaluation metric to understand the semantic difference between COD and SOD and present a baseline method USCNet. In USCNet, the authors considers the SAM com-
ponents including SAM encoder and mask decoder. Along with it, the authors proposes attribute specific prompt generation for saliency, camouflaged, and background prompts.

**Strengths:**

+The authors propose a new dataset CS12K which combines both salient and
camouflaged scenarios.
+The authors present a thorough experimental validation under all the
mentioned scenarios and outperform in all.
+The authors also present a new metric to distinguish between the salient
and camouflaged regions in the image

**Weaknesses:**

- This work tackles only extremes, salient or camouflaged objects. What about intermediate scenarios, for e.g., non-salient or less-salient objects. What about objects with complex backgrounds? If we are trying to generalize, it should cover the entire range, not just the two extremes.

- The scenario of multiple objects seems missing from the dialogue. Although the authors have proposed the dataset by collecting most of the images from previous benchmark datasets, the presence of multiple camouflaged/salient objects (as per real-world scenario) seems to be missing from the dataset or model itself.

- Salient and camouflaged objects belong to different distributions based on their visual appearance and structural complexity. However, the loss function or the network does not cater to that. Is there a way to analyze this difference in the distribution of salient and camouflaged objects so that it does not impact the final prediction?

- Although the authors leverage the SAM-adapter idea, it's strange that they haven't compared with it, which does provide COD results. Also, please provide results with other recent COD and SOD works. Some of them are listed below:
1) CamoFocus [1]
2) CamoDiffusion [2]
3) CamoFormer [3]
4) PGT [4]

- It is unclear how the static prompt queries (SPQ) are generated. Moreover, the operation to predict the coarse operation is also poorly explained. It is hard to see the role of combining dynamic prompt queries (DPQ) and static prompt queries. Moreover, the motivation for the DPQ method is also unclear; authors should explain the reason why a learning component is required for DPQ calculation. Additionally, authors should explain the motivation and how they employ self-attention with salient, camouflaged, and background queries and further generate the prompts for the same.

- In line 355, the authors state that they employ self-attention, while in line 352, they mention cross-attention. This statement is contradictory and unclear.

- In table 5, there seems to be some ambiguity in the results. Specially for ICEG, the results in the manuscript provides different results than what the paper has provided (in all combined settings). Some light on this issue is required.

References:
[1] Abbas Khan, Mustaqeem Khan, Wail Gueaieb, Abdulmotaleb El-Saddik,
Giulia De Masi, and Fakhri Karray. Camofocus: Enhancing camouflage ob-
ject detection with split-feature focal modulation and context refinement. In
IEEE/CVF Winter Conference on Applications of Computer Vision, WACV
2024, Waikoloa, HI, USA, January 3-8, 2024, pages 1423–1432. IEEE, 2024.
[2] Zhongxi Chen, Ke Sun, and Xianming Lin. Camodiffusion: Camouflaged
object detection via conditional diffusion models. In Michael J. Wooldridge,
Jennifer G. Dy, and Sriraam Natarajan, editors, Thirty-Eighth AAAI Confer-
ence on Artificial Intelligence, AAAI 2024, Thirty-Sixth Conference on Innova-
tive Applications of Artificial Intelligence, IAAI 2024, Fourteenth Symposium
on Educational Advances in Artificial Intelligence, EAAI 2014, February 20-27,
2024, Vancouver, Canada, pages 1272–1280. AAAI Press, 2024.
[3] Bowen Yin, Xuying Zhang, Deng-Ping Fan, Shaohui Jiao, Ming-Ming
Cheng, Luc Van Gool, and Qibin Hou. Camoformer: Masked separable atten-
tion for camouflaged object detection. IEEE Transactions on Pattern Analysis
and Machine Intelligence, pages 1–14, 2024.
[4] Rui Wang, Caijuan Shi, Changyu Duan, Weixiang Gao, Hongli Zhu,
Yunchao Wei, and Meiqin Liu. Camouflaged object segmentation with prior via
two-stage training. Comput. Vis. Image Underst., 246:104061, 2024

**Questions:**

Could you please address the issues raised in the weakness section?

---

> ### Author Response · Authors · 2024-11-23
> **Response to Reviewer Htmz (1/5)**
>
> Thank you for your careful reading and the valuable questions you raised. We also appreciate your recognition of our dataset, new metrics, and the performance of our model. In the past few days, we have conducted some experiments on the latest methods and provided detailed explanations addressing your concerns. Below, we provide our detailed responses to your comments and hope that these clarifications address your concerns effectively.
>
> ---
>
> >***Q1:*** This work tackles only extremes, salient or camouflaged objects. What about intermediate scenarios, for e.g., non-salient or less-salient objects. What about objects with complex backgrounds? If we are trying to generalize, it should cover the entire range, not just the two extremes.
>
> ***Ans for Q1:***
>
> **In classic SOD and COD tasks**, such as *[1] [2]*, objects that are neither salient nor camouflaged—i.e., non-salient or less-salient objects, and non-camouflaged or less-camouflaged objects—are typically **treated as background**. These objects either lack distinct visual features or fail to exhibit strong visual similarity to their surroundings. We **maintain this setting**, focusing on distinguishing the salient and camouflaged patterns of objects, while ignoring generic objects.
>
> ---
>
> >***Q2:*** The scenario of multiple objects seems missing from the dialogue. Although the authors have proposed the dataset by collecting most of the images from previous benchmark datasets, the presence of multiple camouflaged/salient objects (as per real-world scenario) seems to be missing from the dataset or model itself.
>
> ***Ans for Q2:***
>
> Our CS12K dataset **contains images with different numbers of objects**. To show it more clearly, we have counted the distribution of images with different numbers of objects in CS12K, as shown in the following *Table R1*.
> Indeed, several previous popular SOD datasets (*DUTS [1]*, *HKU-IS [3]*, and *DUT-OMRON [4]*) and COD datasets (*CAMO [5]*, *COD10K [2]*, and *NC4K [6]*) also contain scenes with multiple camouflaged and/or salient objects.
> Thank you for your suggestions and we will include an analysis of this in the revision.
>
> *Table R1: Distribution of Images with Different Numbers of Objects in CS12K*
> | Number of objects  | 0 | 1 | 2 | >2 |
> | -------- | -------- | -------- | -------- | -------- |
> | Number of images | 3000 | 4197 | 2335 | 2468 |
>
> ---
>
> ---
>
> >***Reference:***
> >
> >*[1]Lijun Wang, Huchuan Lu, Yifan Wang, Mengyang Feng, Dong Wang, Baocai Yin, and Xiang Ruan. Learning to detect salient objects with image-level supervision. In CVPR, 2017a.*
> >
> >*[2]Deng-Ping Fan, Ge-Peng Ji, Guolei Sun, Ming-Ming Cheng, Jianbing Shen, and Ling Shao. Camouflaged object detection. In CVPR, pp. 2777–2787, 2020.*
> >
> >*[3]Guanbin Li and Yizhou Yu. Visual saliency based on multiscale deep features. In CVPR, 2015.*
> >
> >*[4]Chuan Yang, Lihe Zhang, Huchuan Lu, Xiang Ruan, and Ming-Hsuan Yang. Saliency detection via graph-based manifold ranking. In CVPR, 2013.*
> >
> >*[5]Trung-Nghia Le, Tam V Nguyen, Zhongliang Nie, Minh-Triet Tran, and Akihiro Sugimoto. Anabranch network for camouflaged object segmentation. CVIU, 184:45–56, 2019.*
> >
> >*[6]Yunqiu Lv, Jing Zhang, Yuchao Dai, Aixuan Li, Bowen Liu, Nick Barnes, and Deng-Ping Fan. Simultaneously localize, segment and rank the camouflaged objects. In CVPR, pp. 11591–11601, 2021.*

---

> > ### Comment · Reviewer_224k · 2024-11-26
> > **Question**
> >
> > "Ans for Q1:
> >
> > In classic SOD and COD tasks, such as [1] [2], objects that are neither salient nor camouflaged—i.e., non-salient or less-salient objects, and non-camouflaged or less-camouflaged objects—are typically treated as background. These objects either lack distinct visual features or fail to exhibit strong visual similarity to their surroundings. We maintain this setting, focusing on distinguishing the salient and camouflaged patterns of objects, while ignoring generic objects."
> >
> > If focus on these two extremely cases (in opposite direction, salient and camouflaged), why we still need a joint prompt? I think one prompt for salient object detection and the other for camouflaged object detection, like VSCode and previous works. Thus, I am still confused the motivation of this work. Why these two tasks facilitate each other during training?

---

> ### Author Response · Authors · 2024-11-23
> **Response to Reviewer Htmz (2/5)**
>
> >***Q3:*** Salient and camouflaged objects belong to different distributions based on their visual appearance and structural complexity. However, the loss function or the network does not cater to that. Is there a way to analyze this difference in the distribution of salient and camouflaged objects so that it does not impact the final prediction?
>
> ***Ans for Q3:***
>
> As you mentioned, the salient and camouflaged objects belong to different distributions, which is a critical issue. Our network does not independently learn salient and camouflaged objects but instead **models the distributional differences between salient and camouflaged patterns internally**. Specifically, the APG module models the relationships and distinctions among the three attributes (saliency, camouflage, and background), generating three discriminative attribute prompts: $P_S$, $P_C$, and $P_B$ (refer to Fig. 4). This reduces the impact of distributional differences between salient and camouflaged objects on the final prediction. The specific mechanism of APG is detailed in ***Ans for Q5***. Thanks to this, when both salient and camouflaged objects are present in a scene, the network can accurately distinguish and identify them, which is the primary problem we aim to solve.
>
> *Table R2* presents the experimental results of our model with and without the APG module in Scene C (Salient and camouflaged objects coexist) and overall scenes. This results show that our APG module can **simultaneously improve the model's performance in detecting both salient and camouflaged objects across all scenes, with a particular improvement in Scene C**. Compared to the results without APG, the model with APG achieves improvements of 5.79 in $IoU_S↑$ and 7.53 in $IoU_C↑$ in Scene C, and improvements of 4.15 in $IoU_S↑$, 4.63 in $IoU_C↑$, 4.63 in $mIoU↑$, 4.63 in $mAcc↑$, and 1.63 in $CSCS↓$ in Overall Scenes. This indicates that our approach is better adapted to the distributions of salient and camouflaged objects across various scenarios, particularly in scenes where both salient and camouflaged objects coexist.
>
> *Table R2: Effectiveness of APG in Scene C and Overall Scenes*
>
> | Model | APG | Scene C | | Overall Scenes | | | | |
> | -------- | -------- | -------- | -------- | -------- | -------- | -------- | -------- | -------- |
> | | | $IoU_S↑$ | $IoU_C↑$ | $IoU_S↑$ | $IoU_C↑$ | $mIoU↑$ | $mAcc↑$ | $CSCS↓$ |
> | USCNet | ✗ | $69.01$ | $38.20$ | $71.42$ | $56.71$ | $74.98$ | $84.74$ | $9.12$ |
> | USCNet | ✓ | $74.80_{(+5.79)}$ | $45.73_{(+7.53)}$ | $75.57_{(+4.15)}$ | $61.34_{(+4.63)}$ | $78.03_{(+3.05)}$ | $87.92_{(+3.18)}$ | $7.49_{(-1.63)}$ |
>
> To further validate the ability of the APG module in promoting the network's learning of discriminative salient and camouflaged features, we used **Euclidean Distance** to evaluate the feature distribution differences between salient and camouflaged features on the entire CS12K testing set, both with and without using APG. The formula for calculating the Euclidean Distance is as follows:
>
> $Dist = || Fea_s^{(i)} - Fea_c^{(i)} ||^2 = Σ_j ( Fea_s^{(i)}[j] - Fea_c^{(i)}[j] )^2$,
>
> where $Fea_s^{(i)}$ and $Fea_c^{(i)}$ represent the saliency features and camouflage features of the $i$-th sample, respectively, and $j$ is the feature dimension. The $Fea_s$ and $Fea_c$ come from the saliency token and camouflage token generated by the SAM decoder, respectively, where $j$ is the token dimension of 256. We evaluated this on a total of 3,600 images from the CS12K testing set, and the results are as follows:
>
> - **Without APG**, the average Euclidean Distance between salient and camouflaged features is **306.1**.
>
> - **With APG**, the average Euclidean Distance between salient and camouflaged features is **391.7**.
>
> It can be observed that **when using APG**, the Euclidean Distance between saliency and camouflage features in the feature space increases significantly, indicating **a larger spatial separation between them**. This suggests that our method effectively captures the distributional differences between salient and camouflaged objects.

---

> ### Author Response · Authors · 2024-11-23
> **Response to Reviewer Htmz (3/5)**
>
> >***Q4:*** Although the authors leverage the SAM-adapter idea, it's strange that they haven't compared with it, which does provide COD results. Also, please provide results with other recent COD and SOD works. Some of them are listed below:
> >
> > *CamoFocus [1]*
> >
> > *CamoDiffusion [2]*
> >
> > *CamoFormer [3]*
> >
> > *PGT [4]*
>
> ***Ans for Q4:***
>
> In Table 2 of the manuscript, the last row for the COD models shows SAM2-Adapter *[6]*, which is the 2.0 version of SAM-Adapter *[5]* and performs better than SAM-Adapter. To test the upper limit of SAM-Adapter, we evaluated SAM2-Adapter. The experimental results for SAM-Adapter and SAM2-Adapter are shown in *Table R3*. Thank you for the reminder and we will include the SAM-Adapter results in the revision of Table 2.
>
> *Table R3: Quantitative comparisons with SAM-Adapter and SAM2-Adapter.*
> | Model | Venue | Para.(M) | Scene A | Scene B | | Scene C | | | Overall | Scenes | | | |
> | -------- | -------- | -------- | -------- | -------- | -------- | -------- | -------- | -------- | -------- | -------- | -------- | -------- | -------- |
> | | | | $IoU_S$↑ | $IoU_C$↑ | | $IoU_S$↑ $IoU_C$↑ | | | $IoU_S$↑ | $IoU_C$↑ | $mIoU$↑ | $mAcc$↑ | $CSCS$↓ |
> | **SAM-Adapter *[5]*** | ICCVW-23 | 4.11 | 78.90 | 67.69 | | 68.19 27.73 | | | 70.66 | 52.69 | 73.38 | 83.35 | 10.28 |
> | **SAM2-Adapter *[6]*** | Arxiv-24 | 4.36 | 78.75 | 70.28 | | 69.01 38.20 | | | 71.42 | 56.71 | 74.98 | 84.74 | 9.12 |
> | **USCNet (Ours)** | - | **4.04** | **79.70** | **74.99** | | **74.80 45.73** | | | **75.57** | **61.34** | **78.03** | **87.92** | **7.49** |
>
> Regarding the four latest COD works you mentioned, except for CamoFocus *[1]*, which has not yet released its code, we have conducted experiments on the other three COD methods for the USCOD task. After training the three methods on the CS12K training set, we evaluate them on the CS12K testing set, and the experimental results are shown in *Table R4*. We can see that **our proposed USCNet still performs best in all evaluation metrics**. Thank you for providing these new methods and we will include these experimental results in the version of Table 2.
>
> *Table R4: Quantitative comparisons with CamoDiffusion, CamoFormer and PGT.*
> | Model | Venue | Para.(M) | Scene A | Scene B | | Scene C | | | Overall | Scenes | | | |
> | -------- | -------- | -------- | -------- | -------- | -------- | -------- | -------- | -------- | -------- | -------- | -------- | -------- | -------- |
> | | | | $IoU_S$↑ | $IoU_C$↑ | | $IoU_S$↑ $IoU_C$↑ | | | $IoU_S$↑ | $IoU_C$↑ | $mIoU$↑ | $mAcc$↑ | $CSCS$↓ |
> | **CamoDiffusion *[2]*** | AAAI-24 | 72 | 75.01 | 59.39 | | 53.49 45.03 | | | 63.49 | 52.80 | 70.70 | 77.73 | 7.73 |
> | **CamoFormer *[3]*** | TPAMI-24 | 71 | 75.88 | 66.19 | | 73.33 44.14 | | | 71.86 | 56.09 | 74.81 | 84.17 | 7.57 |
> | **PGT *[4]*** | CVIU-24 | 68 | 72.75 | 61.51 | | 70.01 41.21 | | | 71.46 | 56.83 | 75.03 | 83.35 | 9.09 |
> | **USCNet (Ours)** | - | **4.04** | **79.70** | **74.99** | | **74.80 45.73** | | | **75.57** | **61.34** | **78.03** | **87.92** | **7.49** |
>
> ---
>
> ---
>
> >***Reference:***
> >
> >*[1] Abbas Khan, Mustaqeem Khan, Wail Gueaieb, Abdulmotaleb El-Saddik, Giulia De Masi, and Fakhri Karray. Camofocus: Enhancing camouflage ob- ject detection with split-feature focal modulation and context refinement. In IEEE/CVF Winter Conference on Applications of Computer Vision, WACV 2024, Waikoloa, HI, USA, January 3-8, 2024, pages 1423–1432. IEEE, 2024.*
> >
> >*[2] Zhongxi Chen, Ke Sun, and Xianming Lin. Camodiffusion: Camouflaged object detection via conditional diffusion models. In Michael J. Wooldridge, Jennifer G. Dy, and Sriraam Natarajan, editors, Thirty-Eighth AAAI Confer- ence on Artificial Intelligence, AAAI 2024, Thirty-Sixth Conference on Innova- tive Applications of Artificial Intelligence, IAAI 2024, Fourteenth Symposium on Educational Advances in Artificial Intelligence, EAAI 2014, February 20-27, 2024, Vancouver, Canada, pages 1272–1280. AAAI Press, 2024.*
> >
> >*[3] Bowen Yin, Xuying Zhang, Deng-Ping Fan, Shaohui Jiao, Ming-Ming Cheng, Luc Van Gool, and Qibin Hou. Camoformer: Masked separable atten- tion for camouflaged object detection. IEEE Transactions on Pattern Analysis and Machine Intelligence, pages 1–14, 2024.*
> >
> >*[4] Rui Wang, Caijuan Shi, Changyu Duan, Weixiang Gao, Hongli Zhu, Yunchao Wei, and Meiqin Liu. Camouflaged object segmentation with prior via two-stage training. Comput. Vis. Image Underst., 246:104061, 2024*
> >
> >*[5]Tianrun Chen, Lanyun Zhu, Chaotao Deng, Runlong Cao, Yan Wang, Shangzhan Zhang, Zejian Li, Lingyun Sun, Ying Zang, and Papa Mao. Sam-adapter: Adapting segment anything in underperformed scenes. In ICCV, pp. 3367–3375, 2023.*
> >
> >*[6]Tianrun Chen, Ankang Lu, Lanyun Zhu, Chaotao Ding, Chunan Yu, Deyi Ji, Zejian Li, Lingyun Sun, Papa Mao, and Ying Zang. Sam2-adapter: Evaluating & adapting segment anything 2 in downstream tasks: Camouflage, shadow, medical image segmentation, and more. arXiv preprint arXiv:2408.04579, 2024.*

---

> ### Author Response · Authors · 2024-11-23
> **Response to Reviewer Htmz (4/5)**
>
> >***Q5:*** It is unclear how the static prompt queries (SPQ) are generated. Moreover, the operation to predict the coarse operation is also poorly explained. It is hard to see the role of combining dynamic prompt queries (DPQ) and static prompt queries. Moreover, the motivation for the DPQ method is also unclear; authors should explain the reason why a learning component is required for DPQ calculation. Additionally, authors should explain the motivation and how they employ self-attention with salient, camouflaged, and background queries and further generate the prompts for the same.
>
> ***Ans for Q5:***
>
> Our APG module is designed to **extract the discriminative attribute prompts of objects**.
> Our insight is that the detection of salient and camouflaged objects within a sample requires consideration of features **across two dimensions**:
>
> - **Sample-generic features:** For all samples, characteristics such as the size, position, color, and texture of the object serve as important generic features for distinguishing salient and camouflaged objects. These features are applicable in most scenarios and can act as universal criteria for judgment.
>
> - **Sample-specific features:** Relying solely on sample-generic features may not suffice in certain complex situations. For instance, when the salient and camouflaged objects share similar colors or categories, sample-generic features alone are insufficient for effective differentiation. In such cases, it is crucial to consider the specific contextual information within the sample and learn features that are closely associated with the current sample to assist in making an accurate judgment.
>
> By combining these two types of features, we generate more discriminative attribute prompts, enabling more accurate differentiation between salient and camouflaged objects.
>
> Based on this, we incorporate two types of queries in the design of the APG module, each responsible for capturing features at different dimensions:
>
> - **Static Prompt Query (SPQ):** Extracts the sample-generic features, capturing attribute information applicable to all samples.
>
> - **Dynamic Prompt Query (DPQ):** Extracts sample-specific features, focusing on the unique contextual information of the current sample.
>
> The process for obtaining **SPQ**, **DPQ**, and generating the **attribute prompts for saliency, camouflage, and background** is as follows:
>
> 1. **SPQ:** We randomly initialize three sets of learnable queries, namely the static saliency, camouflage, and background prompts queries, denoted as {$Q_{S-S} ∈ R^{N×C}, Q_{S-C} ∈ R^{N×C}, Q_{S-B} ∈ R^{N×C}$}, where $N$ represents the number of queries, and $C$ represents the dimensionality of the queries (kept consistent with the dimensionality of the original SAM prompts).
> These static prompt queries remain unchanged during inference, regardless of the input image, and are designed to capture generic features applicable across all samples.
>
> 2. **DPQ:** We first utilize the obtained **coarse prediction** as an **attention map** to focus on the relevant regions of saliency, camouflage, and background in the image, isolating attribute-specific features from the global image features. Then, a learnable component (refer to the Linear in Fig. 4) is used to downsample the extracted features, aligning them with the dimension of the SPQ. The aligned queries are then treated as DPQ, i.e., the three sets of dynamic prompt queries, representing the dynamic saliency, camouflage, and background prompts queries {$Q_{D-S} ∈ R^{N×C}, Q_{D-C} ∈ R^{N×C}, Q_{D-B} ∈ R^{N×C}$}. These dynamic prompt queries change during inference based on the input image and are used to extract sample-specific features.
>
> 3. **Attribute Prompt:** Combine SPQ and DPQ through a summation operation, resulting in the combined queries {$Q_S ∈ R^{N×C}, Q_C ∈ R^{N×C}, Q_B ∈ R^{N×C}$}. These combined queries are then concatenated to form the attribute prompt query $Q ∈ R^{3×N×C}$. A self-attention is employed to explore the similarities and differences between the queries, learning the relationships among saliency, camouflage, and background features, producing $Q' ∈ R^{3×N×C}$. Subsequently, cross-attention is applied to further enhance the learning among the three attribute queries based on the image features, resulting in $Q'' ∈ R^{3×N×C}$. Finally, the $Q'' ∈ R^{3×N×C}$ serves as the Attribute Prompt, which is separated into the saliency, camouflage, and background prompts {$P_S ∈ R^{N×C}, P_C ∈ R^{N×C}, P_B ∈ R^{N×C}$}.
>
> Thank you for your suggestion. We will provide a more detailed explanation of the working mechanism of APG in the relevant section of the revision.

---

> ### Author Response · Authors · 2024-11-23
> **Response to Reviewer Htmz (5/5)**
>
> >***Q6:*** In line 355, the authors state that they employ self-attention, while in line 352, they mention cross-attention. This statement is contradictory and unclear.
>
> ***Ans for Q6:***
>
> Are you referring to *"cross-attention"* in line 362 and *"self-attention"* in line 355? There should be a comma after *"queries"* in line 355. What we want to express in line 355 is *"we employ self-attention to establish relationships between queries, and query-to-image (Q2I) attention to interact with image embedding."* Thank you for your careful reading. We apologize for any confusion caused by our phrasing, and we will revise this sentence in the revision.
>
> ---
>
> >***Q7:*** In table 5, there seems to be some ambiguity in the results. Specially for ICEG, the results in the manuscript provides different results than what the paper has provided (in all combined settings). Some light on this issue is required.
>
> ***Ans for Q7:***
>
> Table 5 shows the generalization performance of all methods on the testing sets of publicly available COD datasets. Before testing, all models are **trained on our CS12K training set**, which differs from the training set used in the original ICEG paper. Therefore, the results in the manuscript differ from those in the original paper. The training data is discussed in L860 of the manuscript.
>
> Additionally, the ICEG backbone in Table 5 of the manuscript is **ResNet50**, which is the **common setting in the original paper**. We also trained ICEG with other backbones from the original paper on our dataset, evaluated it on the CS12K testing set, and conducted generalization experiments on popular SOD and COD datasets. The results of the comparison experiments are shown in *Table R5*, and the results of the generalization experiments are shown in *Table R6* and *Table R7*.
>
> *Table R5: Performance of ICEG with diverse backbones on the CS12K testing set.*
> | Model | Backbone | Scene A | Scene B | | Scene C | | | Overall | Scenes | | | |
> | -------- | -------- | -------- | -------- | -------- | -------- | -------- | -------- | -------- | -------- | -------- | -------- | -------- |
> | | | $IoU_S$↑ | $IoU_C$↑ | | $IoU_S$↑ $IoU_C$↑ | | | $IoU_S$↑ | $IoU_C$↑ | $mIoU$↑ | $mAcc$↑ | $CSCS$↓ |
> | **ICEG** | ResNet50 | 68.28 |  52.21 | |  64.86 34.29 | | |  62.72 |  45.52 |  67.65 |  78.00 | 8.61 |
> | **ICEG** | Res2Net50 | 68.40 | 53.36 | | 64.55 34.76 | | | 63.91 | 45.28 | 68.26 | 77.91 | 8.76 |
> | **ICEG** | Swin | **73.67** | **68.38** | | **68.43** **44.33** | | | **69.22** | **58.71** | **74.68** | **83.53** | **8.16** |
>
> *Table R6: Performance of ICEG with diverse backbones on the DUTS, HKU-IS, and DUT-OMRON testing sets. DUT-O denotes DUT-OMRON.*
> | Model | Backbone | DUTS | | | | | | HKU-IS | | | | | | DUT-O | | | | |
> | -------- | -------- | -------- | -------- | -------- | -------- | -------- | -------- | -------- | -------- | -------- | -------- | -------- | -------- | -------- | -------- | -------- | -------- | -------- |
> | | | $F_β^{max}$↑ | $F_β^{ω}$↑ | $M$↓ | $S_α$↑ | $E_ϕ^m$↑ | | $F_β^{max}$↑ | $F_β^{ω}$↑ | $M$↓ | $S_α$↑ | $E_ϕ^m$↑ | | $F_β^{max}$↑ | $F_β^{ω}$↑ | $M$↓ | $S_α$↑ | $E_ϕ^m$↑ |
> | **ICEG** | ResNet50 | .679 | .647 | .063 | .756 | .807 | | .803 | .773 | .056 | .819 | .882 | | .603 | .570 | .084 | .712 | .751 |
> | **ICEG** | Res2Net50 | .664 | .640 | .063 | .750 | .782 | | .792 | .770 | .056 | .816 | .869 | | .596 | .571 | .081 | .713 | .736 |
> | **ICEG** | Swin | **.719** | **.700** | **.050** | **.789** | **.820** | | **.832** | **.815** | **.045** | **.848** | **.896** | | **.664** | **.645** | **.061** | **.762** | **.785** |
>
> *Table R7: Performance of ICEG with diverse backbones on the CAMO, COD10K, and NC4K testing sets.*
> | Model | Backbone | CAMO | | | | | | NC4K | | | | | | COD10K | | | | |
> | -------- | -------- | -------- | -------- | -------- | -------- | -------- | -------- | -------- | -------- | -------- | -------- | -------- | -------- | -------- | -------- | -------- | -------- | -------- |
> | | | $F_β^{max}$↑ | $F_β^{ω}$↑ | $M$↓ | $S_α$↑ | $E_ϕ^m$↑ | | $F_β^{max}$↑ | $F_β^{ω}$↑ | $M$↓ | $S_α$↑ | $E_ϕ^m$↑ | | $F_β^{max}$↑ | $F_β^{ω}$↑ | $M$↓ | $S_α$↑ | $E_ϕ^m$↑ |
> | **ICEG** | ResNet50 | .560 | .525 | .113 | .650 | .687 | | .551 | .513 | .048 | .689 | .729 | | .623 | .592 | .075 | .705 | .746 |
> | **ICEG** | Res2Net50 |.560 | .524 | .114 | .656 | .681 | | .643 | .613 | .071 | .725 | .767 | | .552 | .517 | .048 | .695 | .732 |
> | **ICEG** | Swin | **.728** | **.697** | **.066** | **.769** | **.820** | | **.735** | **.708** | **.051** | **.786** | **.840** | | **.645** | **.610** | **.035** | **.753** | **.807** |

---

> ### Author Response · Authors · 2024-11-29
> **Response to Reviewer 224k**
>
> Thank you for your feedback. We hope our response can address your concerns.
>
> ---
>
> Although salient and camouflaged objects are in opposing directions, establishing a connection between them during training is still necessary. In fact, **both VSCode and our method establish a connection between prompts**, but the way the connection is formed differs due to the different focuses of the models. Specifically, **VSCode** uses **two separate prompts** to handle the **SOD and COD tasks separately**, but these prompts are not completely independent in their learning. It uses **discriminative loss** to reduce the similarity between different prompts, thereby **indirectly** establishing a connection to learn the feature differences between salient and camouflaged objects **across different samples (inter-image samples)**, effectively handling seven different multimodal SOD and COD tasks. In contrast, our **joint prompt**, designed for **USCOD**, **directly** models the interaction between prompts within the network through the APG module. This approach is more suited for learning the feature distribution of salient and camouflaged objects **within samples (intra-image samples)**.
>
> In the USCOD task, the joint prompt has two advantages:
>
> - **Adaptation to Complex Scenes**: Salient and camouflaged objects are not merely opposites in terms of similarity, especially when they coexist in the same scene. For example, camouflaged objects may be hidden within salient objects (e.g., flowers and insects of the same color) or may belong to the same species with similar morphological features, where factors like lighting or background cause one to appear salient and the other camouflaged. In such complex scenes, salient and camouflaged objects **may share certain features**, and simply reducing prompt similarity may not suffice to capture the intricate relationship between them. Our APG module directly models the interaction between prompts, enabling the model to **flexibly adapt to the specific feature distribution of the scene**, rather than being limited to simple similarity measures. This approach allows for better handling of the complex and diverse scenarios in the USCOD task.
>
> - **Avoiding Pixel Uncertainty**: VSCode has demonstrated strong generalization across seven different multimodal SOD and COD tasks, but it primarily targets scenes with only single attribute objects (Scene A and B in the manuscript). When aiming to detect salient and camouflaged objects in Scene C (salient and camouflaged objects coexist), VSCode needs to separately use the SOD and COD prompts to handle the same image, resulting in two independent masks. This approach often leads to overlapping pixel regions in the detection results. For example, *in Figure 4 of the VSCode paper*, when using the SOD prompt, the network detects the salient palace and part of the camouflaged person, while using the COD prompt, the camouflaged person is detected again, resulting in **overlapping pixels and increased detection uncertainty**. In contrast, our joint prompts directly model the relationship between salient and camouflaged objects through the APG module, **automatically learning uncertain pixels** within the same scene without the need for post-processing. This approach not only alleviates the overlapping pixel issue but also enhances the stability and accuracy of the detection results.
>
> Overall, **VSCode** demonstrates strong generalization across **seven different multimodal SOD and COD tasks**, effectively learning the feature differences between salient and camouflaged objects **across different samples (inter-image samples)** by **indirectly** establishing connections between prompts. In contrast, **our model** focuses on the **USCOD task** and uses joint prompts to **directly** model the interaction between prompts, enabling mutual learning between the two attributes during training. This direct interaction approach is better adapted to the feature distribution of salient and camouflaged objects **within the same sample (intra-image sample)**, effectively improving the model's performance on the USCOD task.
>
> ---
>
> If you have any further questions or need additional clarification, please feel free to let us know, and we will respond promptly.

---

> ### Author Response · Authors · 2024-12-02
> **Last Day Remainder to Reviewer Htmz**
>
> Dear Reviewer Htmz,
>
> We have provided responses and additional experimental results for your concerns, and we hope that we have addressed most of your questions. Since this is toward the end of the rebuttal, please don't hesitate to let us know if you have any final questions or comments. Thank you!
>
> Best regards,
>
> Authors of #38

---

> ### Comment · Reviewer_Htmz · 2024-12-02
> **Response to rebuttal (part 1)**
>
> Dear Authors,
>
> Thanks for the rebuttal.
>
> However, my fundamental question on covering the range between salient and camouflaged still remains.
> If you are saying less salient or less camouflaged objects should be treated as background, then all the research done over the years on foreground extraction in complex scenes appears meaningless.
>
> My viewpoint is the following: If you are covering extremes, you should cover everything in-between as well, and call it generic foreground extraction instead. You should be aiming for foreground extraction regardless of whether it's salient, less salient, less camouflaged or camouflaged. I just don't understand what purpose does it serve by having a joint network for the two extremes. Can you think of any real-world application/setting for this?

---

> ### Author Response · Authors · 2024-12-03
> **Response to Reviewer Htmz**
>
> Thank you for your feedback. We hope our response can address your concerns.
>
> ---
>
> We are not dismissing prior research on foreground extraction.  **This paper does not aim to overturn the task setting of SOD and COD but rather extends them.**  Previous SOD and COD methods assume that a salient object and a camouflaged object will not appear in an image simultaneously, and learning saliency and camouflage **inter-image samples** (scene A and B in the paper), while our work introduces a more relaxed setting: enabling the model to simultaneously learn saliency and camouflage **intra-image sample** (scene C in the paper), thus making the model more unconstrained.
>
> The general foreground extraction you mentioned is typically more generalized, aiming only to separate foreground from background without distinguishing the specific attributes of objects within the foreground. The key difference in our approach is that we shift the region of interest from the entire foreground to salient and camouflaged objects, explicitly assigning these attributes.
>
> In practical applications, the two tasks have different focus areas:
>
> - **General foreground extraction** is applied to the foreground detection needs in generic scenarios, such as image editing, video surveillance, and video monitoring, without considering the saliency or camouflage of the objects.
>
> - **The USCOD task** is more focused on special scenario analysis, where it plays a crucial role in improving operational efficiency (such as post-disaster rescue, multi-level lesion detection) and enhancing safety (such as diving hazard warnings, military surveillance). In these scenarios, salient objects are often considered a higher priority, prompting quicker decision-making or action, while camouflaged objects are seen as potential risks, prompting targeted actions to address them. Simultaneously detecting both enhances decision-making comprehensiveness and effectiveness.
>
> **The typical real-world applications of USCOD are as follows:**
>
> - **Military Surveillance and Enemy Reconnaissance**: In a military environment, salient objects could be large military equipment (such as tanks, helicopters), which are usually clear sources of threat and require immediate attention and action. Camouflaged objects could be soldiers or equipment hidden in vegetation or camouflage materials, often overlooked but potentially carrying deadly threats. Salient objects require quick identification and military strikes to weaken enemy combat capabilities, while camouflaged objects, as hidden threats, need extra attention to prevent oversight. Simultaneously detecting both can balance addressing direct threats and eliminating potential ones, thereby enhancing battlefield situational awareness.
>
> - **Post-Disaster Search and Rescue**: After a disaster, salient objects could be clear signs of life in the rubble (such as visibly trapped individuals), representing the immediate focus for rescue efforts, where quick identification and action can save lives. Camouflaged objects might be life signs concealed due to obstruction or chaotic environments (such as partially buried survivors), often overlooked and more difficult to rescue. Prioritizing the rescue of salient objects increases efficiency, while camouflaged objects might require more manpower and resources to reach. Simultaneously detecting both helps allocate resources effectively and ensure accurate rescue orders.
>
> - **Multi-Level Lesion Detection**: In medical imaging, salient objects might include obvious tumors or organ damage, typically the first focus for doctors. Camouflaged objects could be lesions with blurred boundaries or early-stage abnormalities, which may indicate early-stage diseases or more complex issues. Detecting salient lesions allows doctors to quickly determine treatment priorities, while identifying camouflaged lesions helps catch potential health problems early, preventing disease progression. Simultaneously detecting both improves diagnostic efficiency and accuracy, helping to develop the best treatment plan for patients.
>
> - **Diving Hazard Warnings**: During a dive, salient objects might include coral or schools of fish that attract divers' attention, contributing to an enriching visual experience. Camouflaged objects could be dangerous creatures like stonefish or moray eels, whose concealment increases potential risks, and failure to detect them could jeopardize the diver's safety. Detecting salient objects can guide divers to interesting spots, enhancing the diving experience, while identifying camouflaged objects helps divers avoid potential dangers and ensure safety. Simultaneously detecting both improves the diving experience and reduces safety risks.
>
> These applications are also discussed *in §G of the Appendix in the revision*. If you have any further concerns, we will do our best to address them.

---

### Author Response · Authors · 2024-11-29
**Gratitude for Reviewers' Feedback and Summary of Contributions**

Dear AC and Reviewers,

We would like to express our sincere gratitude for your valuable feedback and insightful comments on our paper. Below, we provide a summary of our contributions and a brief acknowledgment of the reviewers’ recognition.

**Our Contributions:**
1. **New Benchmark: USCOD**
   We propose a new benchmark called USCOD, designed to detect both salient and camouflaged objects in unconstrained scenes. This benchmark addresses the limitations of traditional Salient Object Detection (SOD) and Camouflaged Object Detection (COD) tasks, which generally focus on specific types of scenes.

2. **New Dataset: CS12K**
   We introduce CS12K, a large-scale dataset for USCOD, consisting of four diverse scenes with both mask-level and attribute annotations. To our knowledge, CS12K is the first dataset to cover a variety of scenes without restricting the presence of salient or camouflaged objects, offering a more comprehensive challenge for current detection methods.

3. **New Evaluation Metric: CSCS**
   We propose the Camouflage-Saliency Confusion Score (CSCS), a new metric that evaluates the confusion between camouflaged and salient objects. This fills a crucial gap in existing evaluation methods, which typically do not address the unique challenges presented by scenes containing both object types.

4. **Proposed Baseline for USCOD: USCNet**
   We introduce USCNet, a baseline method for USCOD that combines Static Prompt Query (SPQ) and Dynamic Prompt Query (DPQ) to query both generic and sample-specific features. This allows USCNet to generate discriminative attribute-specific prompts for detecting both salient and camouflaged objects, which achieves the new state-of-the-art performance on USCOD task.

5. **Extensive Experiments and Comparative Analysis**
   We conduct extensive experiments comparing USCNet with 19 existing SOD and COD methods on the CS12K dataset across four different scenes, providing a thorough analysis of performance gaps and challenges within USCOD.

**Summary:**

Our work introduces a novel benchmark, dataset, evaluation metric, and baseline method for USCOD. We believe these contributions can advance the field of salient and camouflaged object detection, paving the way for **future research** in these areas. Additionally, this work holds significant potential for **application** in special scenario analysis, where it plays a crucial role in improving operational efficiency and enhancing safety.

**Acknowledgments:**

- **Contribution of CS12K Dataset:** We sincerely thank **all reviewers** for recognizing the significant contribution of the CS12K dataset to the field of USCOD research.

- **Value of the CSCS Metric:** We are grateful to **Reviewer Htmz and Reviewer FwYu** for acknowledging the value of the CSCS as an essential evaluation metric in handling scenes with both types of objects.

- **Validation of Our Approach:** Thanks to **Reviewer Htmz, Reviewer 224k, and Reviewer LUqD** for acknowledging the validity of our approach, which has been thoroughly verified through extensive experiments. We also appreciate their recognition of the necessity for designing specialized datasets and networks for USCOD.

We deeply appreciate your support and constructive feedback, which has greatly contributed to the improvement of our work.

Thank you for your time and consideration.

Best regards,

Authors of #38

---

### Meta-Review · Area_Chair_fW1R · 2024-12-21

**Metareview:**

In this paper, the authors presented a new task -- unconstrained salient and camouflaged object detection (USCOD), that linked the conventional tasks of visual salient object detection (SOD) and camouflaged object detection (COD). Specifically, a USCOD benchmark was introduced, aiming at the simultaneous detection of salient and camouflaged objects. A large-scale dataset (CS12K) was constructed to support this. A baseline model was presented to address this problem. Experimental results show the effectiveness of the proposed method on this new USCOD task. The main strengths of this paper are:
- A new USCOD benchmark was introduced, with a supported dataset CS12K and a corresponding new evaluation metric. The large-scale dataset could be a good contribution to either the salient object detection community, the camouflaged object detection community, or the newly introduced USCOD task.
- A thorough experimental evaluation was presented, showing the effectiveness of the proposed method.
- The paper is generally well-written and easy to follow.

The main weaknesses of this paper include:
The definition, motivation, and rationale of the introduced USCOD task are questionable. This is a major concern shared by the reviewers and AC. The designed USCOD task was considered to tackle only extremes, salient or camouflaged objects, with an unclear practical application scenario. The corresponding technical design was also unclear.
- The novelty and contribution of the proposed method were found to be limited. The technical design of the components proposed was not that novel, given similar designs have been proposed in prior works (more details please see the review comments). Whereas the authors failed to provide a detailed convincing clarification and comparison for this.
- Some of the technical designs and experimental settings lack information, and some statements are unclear without convincing evidence to support them. For more details, please refer to the review comments.

Although some minor concerns were addressed by the authors through the rebuttal and discussions, the major concerns (especially the first two mentioned above) remain. Even after reading the authors' explanation, the reviewers and AC are still not convinced by the designed new USCOD task, there is a considerable disagreement over this. This new task is more like an artificially designed task rather than a real challenging scenario/problem that was well motivated. Before clearing this key issue, the paper in its current form is considered to be not ready to present at ICLR. Apart from this, the novelty concern also stands. The AC (and reviewers) understand and appreciate the good contribution of the new large-scale dataset this paper brings to the community, but considering the major concern about the key problem this paper studied, the AC agreed with the reviewers that this paper is not ready to be presented at ICLR, and encourage the authors to carefully consider the reviewers' comments (especially the key concern mentioned above) to revise their paper.

**Additional Comments On Reviewer Discussion:**

This paper received review comments from 4 expert reviewers. During the rebuttal period, there was a heated discussion between the authors and reviewers with good engagement. Through this authors-reviewers discussion phase, some concerns were well addressed, but there are still a few major issues remaining (see above). There was reviewer increased their rating and also reviewer decreased their rating, ending up with 2 Reject, 1 Borderline Reject, and 1 Accept. Overall the rating was negative, but this was not the sole reason that led to the final decision. The main reasoning behind the decision was mainly due to the key issue of the questionable definition and settings of the newly introduced task.

The AC understand that this decision might be discouraging and the authors may not agree with it, but hope the comments and discussions from the reviewers could be helpful for improving their paper for potential future submission.

---

### Decision · Program_Chairs · 2025-01-22

Reject